# Application of Single Particle ICP-MS for the Determination of Inorganic Nanoparticles in Food Additives and Food: A Short Review

**DOI:** 10.3390/nano13182547

**Published:** 2023-09-12

**Authors:** Katrin Loeschner, Monique E. Johnson, Antonio R. Montoro Bustos

**Affiliations:** 1Research Group for Analytical Food Chemistry, National Food Institute, Technical University of Denmark, 2800 Kgs. Lyngby, Denmark; 2Material Measurement Laboratory, Chemical Sciences Division, National Institute of Standards and Technology (NIST), Gaithersburg, MD 20899, USA; monique.johnson@nist.gov (M.E.J.); antonio.montorobustos@nist.gov (A.R.M.B.)

**Keywords:** inorganic nanoparticles, single particle ICP-MS, food, food additives, sample collection, sample preparation, method validation

## Abstract

Due to enhanced properties at the nanoscale, nanomaterials (NMs) have been incorporated into foods, food additives, and food packaging materials. Knowledge gaps related to (but not limited to) fate, transport, bioaccumulation, and toxicity of nanomaterials have led to an expedient need to expand research efforts in the food research field. While classical techniques can provide information on dilute suspensions, these techniques sample a low throughput of nanoparticles (NPs) in the suspension and are limited in the range of the measurement metrics so orthogonal techniques must be used in tandem to fill in measurement gaps. New and innovative characterization techniques have been developed and optimized for employment in food nano-characterization. Single particle inductively coupled plasma mass spectrometry, a high-throughput nanoparticle characterization technique capable of providing vital measurands of NP-containing samples such as size distribution, number concentration, and NP evolution has been employed as a characterization technique in food research since its inception. Here, we offer a short, critical review highlighting existing studies that employ spICP-MS in food research with a particular focus on method validation and trends in sample preparation and spICP-MS methodology. Importantly, we identify and address areas in research as well as offer insights into yet to be addressed knowledge gaps in methodology.

## 1. Introduction

Food is one major source of inorganic nanoparticle (NPs) exposure to consumers via the oral route/ingestion [1]. NPs that are potentially present in foods are naturally occurring or from anthropogenic origins. The latter can be divided into engineered and incidental NPs. Engineered NPs might be intentionally added to food. Currently, no engineered NPs are approved for addition to food in Europe; however, the first novel food in nanoparticulate form, iron hydroxide adipate tartrate, was recently evaluated as safe by the European Food Safety Authority [2]. Engineered NPs have the potential of release from food contact materials or can enter the food chain via the environment when they are used in other applications, e.g., construction and buildings. Incidental NPs might be formed and released during the preparation or production of food. In addition, several approved food additives such as the color additive forms of silver [3] and titanium dioxide (TiO_2_) [4] can release or contain a fraction of particles at the nanoscale. Despite almost three decades of intense research into the toxicology of engineered nanomaterials (NMs), the understanding of their direct impact on human health is still limited [5]. It is expected that, following oral exposure, the largest fraction of ingested NPs quickly passes through the gastrointestinal tract and is lost via fecal matter with a typical translocation to distal organs of less than 1% [5]. Due to the lack of suitable studies, it is not yet possible to conclude or even rank the toxicity of different types of NPs following oral exposure [5]. Many knowledge gaps still exist including the direct effects of NPs in food on gastrointestinal tissues and microbiota within the gastrointestinal tract [6].

The United States Food and Drug Administration (US FDA), e.g., regulates a wide range of US products including food, cosmetic products, drugs and drug formulations, devices, veterinary products, and tobacco products that may utilize nanotechnology in production or contain NMs [7]. More specifically, the incorporation of NMs into food and cosmetics is regulated by the Center for Food Safety and Applied Nutrition (CFSAN). This center is focused on improving information regarding the safety assessments for NMs in order to inform regulatory decision-making. While information is still being gathered by the US FDA to make regulatory decisions regarding safety, research has been utilized to inform uses and restrictions, and the allowable quantities of a given substance (in weight percent) when incorporated as a food additive, as well as recommendations for labeling. All this information is kept in an up-to-date database designated as the Code of Federal Regulations under Title 21—Food and Drugs [8].

The European Food Safety Authority (EFSA) has, e.g., developed specific guidelines regarding the risk assessment of NMs and small particles to be applied in the food and feed chain [9,10]. In the ongoing re-evaluation and follow-up activities regarding the safety of permitted food additives by EFSA, the presence of small particles is considered and the conventional risk assessment, if necessary, is complemented with nano-specific considerations. For example, a detailed risk assessment was performed for the food additive titanium dioxide (E171). A concern for genotoxicity could not be ruled out, and given many uncertainties, the EFSA Panel concluded that E171 could no longer be considered as safe when used as a food additive [11]. The aforementioned discussions and the following ban in Europe initiated several activities toward the analysis of E171 in foods.

Countries worldwide have examined the suitability of their regulatory frameworks for dealing with nanotechnologies in the agricultural, feed, and food sectors. The European Union (EU), along with Switzerland, was identified by Amenta et al. as the only world region where nano-specific provisions have been incorporated in existing legislation in 2015 [12]. In other regions, nanomaterials were regulated more implicitly by mainly building on guidance for industry [12]. Labeling for the presence of NMs as ingredients in food is, e.g., mandatory in the EU since December 2014 in accordance with Regulation No. 1169/2011 [12]. Regulation 1169/2011 states that all ingredients present in the form of engineered NMs shall be clearly indicated in the list of ingredients and that the names of such ingredients shall be followed by the word “nano” in brackets. EngineeredNMs are considered a novel food (if they had not been used for human consumption to a significant degree within the EU before 15 May 1997) and are covered by the novel food regulation [13]. Specific provisions for their safety assessment and authorization as food apply from 2018 onwards [13].

In the contexts of risk assessment, food labeling, and the development of novel foods, reliable detection and characterization methods for NPs in foods are needed. Studies are required to determine the level of NPs in food to allow an assessment of consumer exposure. For food control purposes, it is necessary to know whether intentionally added engineered NPs and food additives containing small particles can be distinguished from the background level of natural and incidental NPs [14].

Several analytical techniques for the characterization of NPs in food exist. The ones most frequently applied are electron microscopy (EM), asymmetric flow field-flow fractionation (AF4) coupled to inductively coupled plasma-mass spectrometry (ICP-MS), and ICP-MS in single particle mode (spICP-MS) [15,16]. Electron microscopy is considered the gold standard for the determination of particle sizes and provides additional information on particle shape, aggregation state, crystal structure, and if combined with spectroscopy, chemical composition. However, it requires special instrumentation that is not typically used in food control laboratories. AF4 faces reproducibility issues (mainly because of particle losses on the membrane) and requires experienced operators [15,16]. Single particle ICP-MS is a promising technique for the screening of food samples for the presence of metal-containing NPs, as it provides information on particle size and particle number concentration with high sensitivity and elemental specificity. Further advantages of spICP-MS are fast analysis, relatively simple sample pre-treatment, and easy implementation in state-of-the-art ICP−MS instruments, which otherwise can be used for metal analysis and speciation. There have been several reviews focusing on the topic of spICP-MS [17,18,19,20,21] discussing its principles, potential, benefits, limitations, and selected applications.

Sample preparation is a very critical step when it comes to the spICP-MS analysis of NPs in food. When using conventional sample introduction systems, i.e., pneumatic nebulizers, combined with a spray chamber, aqueous suspensions of the NPs are required. This means that the matrix of semi-solid or solid foods needs to be degraded. This can be achieved by acidic, alkaline, or enzymatic digestion. However, changes of the NPs, in particular dissolution and agglomeration, need to be avoided. For this reason, acid digestion (as used classically for elemental analysis) is usually not applied. When the aim of the analysis is to study the size of the constituent particles, the highest possible degree of de-agglomeration is desired, as spICP-MS cannot distinguish between individual particles and particles in an agglomerated state. De-agglomeration might, e.g., be achieved by applying probe sonication. However, sonication probes can release particles especially after a certain time of operation due to erosion. This might be problematic if the probe material contains the same element as the NPs present in the samples. An example is Ti-containing NPs which can be released from probes made of titanium alloy. Contamination can be reduced by using new/visually undamaged probes and monitored using procedural blanks. Another alternative is the use of indirect sonication using cup horns, vial tweeters, and similar devices.

Further, clogging of the instrument’s sample introduction system, especially the nebulizer, needs to be prevented by removing any large matrix components/residues of the matrix degradation by filtration, settling (sedimentation), or centrifugation.

A systematic literature review highlighting the determination of metallic NPs in biological samples by spICP-MS was recently performed by Laycock et al. [22]. The review included the organs or tissues collected from animals, plant tissues, and body fluids. The authors identified 83 relevant papers, with the latest search conducted in January 2021. The aim of this short review is to give an overview of existing studies where spICP-MS is used to study NPs in food additives, food, and food-relevant matrices and to identify knowledge gaps for future research.

## 2. Literature Search

The literature search was performed in the Web of Science database, accessed on 30 May 2023. The search commands included the terms: “SP-ICP-MS” or “SP-ICPMS” or “sp-ICPMS” or “spICPMS” or “single particle ICPMS” or “single particle ICP-MS” or “single particle inductively coupled plasma mass spectrometry” or “single particle inductively coupled plasma mass-spectrometry” and “food”, “feed”, “food additive”, “food matrix” (the characters are not case sensitive). The search was then extended to the terms: “plants”, “animals”, “tissues”, “bread”, “biscuity”, “cereal”, “pastry”, “oil”, “nut”, “margarine”, “mayonnaise”, “vinaigrette”, “salad dressing”, “egg”, “milk”, “cream”, “butter”, “yogurt”, “cheese”, “whey”, “casein”, “fruit”, “vegetables”, “legumes”, “seafood”, “seaweed”, “fish”, “mussels”, “shellfish”, “crustaceans”, “meat”, “offal”, “soft drinks”, “beverages”, “soda”, “juice”, “coffee”, “tea”, “beer”, “wine”, “bottled water”, “candy”, “confectionery”, “chewing gum”, “E171”, “E 171”, “E172”, “E 172”, “E173”, “E 173”, “E174”, “E 174”, “E551”, “E 551”. A total of 49 papers were considered relevant and will be thoroughly discussed in this short review article.

This review excluded studies of biological samples that are not typically consumed by humans. These include tissues from humans, rats, mice, whales, earthworms, amphipods, soil nematodes, etc., as well as fish intestines, liver, gills, and brains. Regarding studies on plant tissues, this review focuses on plants that are a part of the typical diet and their eatable parts. Roots of wheat plants and leaves of tomato plants, e.g., were not considered. Garden cress (*Lepidium sativum*) which is used for culinary seasoning is included but thale cress/mouse-ear cress (*Arabidopsis thaliana*) which is a model organism in plant research but without any culinary use is excluded.

It was further decided to exclude papers dealing with the analysis of NPs in all types of waters and only include drinks and beverages.

## 3. Studied Food Matrices and Nanoparticles

Table 1 presents an overview of the sample characteristics, sampling, and sample preparation addressed in these papers. It was decided to divide the studies based on the characteristics of the food matrix as this largely impacts sample preparation.

Many of the papers focus on protein-rich foods (Figure 1A), where the main food type is seafood (shellfish and fish), followed by sugar-rich foods (chewing gum, candies, cake decorations, and similar) and starch-/dietary fiber-rich foods. The last category is mainly comprised of leaf vegetables but also radish and seaweed. There are almost no studies on foods rich in starch, with the exceptions of wheat flour and noodles. The number of studies on fat-rich foods and emulsions is rather limited and includes salad dressing and sour cream.

The most frequently studied particle types were silver- and titanium-containing NPs (Figure 1B). The composition of NPs studied was rarely confirmed but certain assumptions were made, mainly based on the origin of the samples; whereas AuNPs and AgNPs typically contained no other elements, detected Ti was typically assumed to be TiO_2_, Ce to be CeO_2_, and Zn to be ZnO. Al, Cu, and Fe can be in NPs of various compositions. The influence of choosing a certain NP composition on NP diameter and NP mass concentration was, e.g., discussed by Vidmar et al. [14].

Screening studies looking for different elements in one food matrix are currently limited and include the investigation of Al-, Ag-, and Au-containing NPs in powders for decoration of confectionery and coated candy beads (all containing the respective food additives) [27], Ag-, Ti-, Cu-, and Zn- bearing NPs in diverse marine bivalve mollusks [52], 20 selected elements for NPs analysis in clams and oysters of which only six rare earth elements (Y, La, Ce, Pr, Nd, Gd) were detected [54] and eight elements (Ag, Al, Cr, Cu, Fe, Si, Ti, Zn) in 13 food products [14].

All presented studies assume spherical NPs when calculating particle size. Many publications (17 out of 49) report that the studied NPs were in fact spherical or near/almost spherical shaped (Figure 1C). Approximately half of the papers did not provide information highlighting NP shape. In some cases, electron microscopy images were presented but the shapes were neither described nor discussed. Some studies reported clear deviations from a spherical shape, including irregular, ellipsoidal, polygonal, rod-, and flake-like shapes as well as fractal aggregate structures. The relation between the obtained sizes by spICP-MS and the geometric size of the NPs was not always discussed.

The origin of the studied NPs was dominated by the NPs present in food additives, followed by spiking experiments and exposure studies with engineered NPs (Figure 1D). Studies on seafood/aquatic organisms most often referred to the presence of NPs in the aquatic environment, which can be either naturally occurring or anthropogenic NPs (including incidental and engineered NPs) [51]. Otherwise, studies focusing on naturally occurring NPs in food were missing. More “exotic cases” were biogenically formed NPs and included selenium NPs in bacterial strains used in the dairy industry [29] and mercury selenide NPs in fish and mollusks [67], lead NPs in game meat following the use of lead-containing bullets [64] and zinc oxide NPs migrating from a food contact material into chicken breast [63]. In some studies, the origin of the NPs was unknown but possible sources were discussed [14,39].

Thus far, the following food additives were studied: titanium dioxide (E171), iron oxides and hydroxides (E172), aluminum (E173), silver (E174), gold (E175), silicon dioxide (E551), and mica (potassium aluminum silicate (E555) with the focus being on titanium dioxide (13 articles). Titanium dioxide (E171) was studied as pure food additive but also incorporated in candies and chewing gums (coating), crab sticks, and salad dressing as a color. Food additive silver (E174) was investigated in five publications both as a pure additive and in candies/pastry decoration. Food additive gold (E175) was only studied in one paper where no NPs could be detected [27]. The absence of NPs in food additive gold was confirmed in a scientific report to the EFSA [69].

For iron oxides and hydroxides (E172), aluminum (E173), silicon dioxide (E551), and mica (E555), only one paper each exists [14] (E172 and E551), [27] (E173 and E555). None of the papers presented results related to method validation, as will be discussed in detail in Section 6.

## 4. Sample Collection and Sample Preparation

As seen in Figure 2A, a majority of the samples in the studies we highlight here were procured from local stores and markets, with a small size of samples originating from national control programs and online stores. In studies where sugar-rich foods such as candies were analyzed, sample sizes varied greatly; either single candy pieces from a package or batch (12 studies) were prepared or candies were pooled together to reach a critical mass of sample (eight studies). For vegetative samples, common practices included analysis of 1 g of dry leaves or ground plant tissues; however, there were also instances where individual leaves were cut into smaller squares. For meat products, 0.5 g samples were the preferred and most prevalently used sample weight for sample treatment and subsequent analysis. With regard to seafood samples, either dry or wet homogenized samples of 0.5 g or 1 g were prepared and analyzed. As far as the pure food additives themselves, sample sizes as small as 3.5 mg were used as these are typically rich in metal content.

Regarding homogenization approaches, there is no standard, agreed-upon protocol among the studies. Physical homogenization techniques were rarely employed for candy-type samples, but efforts of pooling samples were often used (i.e., pooling of chewing gum samples). Vegetative sample types underwent manual grinding and/or probe sonication procedures. In isolated studies, meat product samples underwent cryo-milling protocols to produce a slurry mix. Seafood samples underwent mechanical blending of large masses of sample (1 kg) and in some cases freeze-drying was employed. Overall, the homogenization procedures utilized in the studies herein were highly unreported among all of the research papers.

No matrix degradation (Figure 2B) was used in the case of pure food additives and liquid-based food such as drinks. In the latter case, simple dilution of the samples is usually applied. Dissolution of the matrix, typically in ultrapure water, was commonly applied for sugar-rich foods. Dispersion of the matrix with the help of sonication was only used in a few cases, e.g., for a salad dressing. NPs in protein-rich foods such as seafood and meat were either extracted by alkaline digestion with tetramethylammonium hydroxide (TMAH) or enzymatic digestion. Proteinase K, a broad-spectrum serine protease, was often used for protein-rich foods such as meat, milk, and seafood. A mixture of pancreatin and lipase has been applied to seafood in several works due to its fat content. An extensive comparison of existing enzymatic and alkaline digestion protocols for bivalve mollusks was performed by Sun et al. [51] which concluded that the optimal extraction was based on the employment of TMAH. Enzymatic digestion with Macerozyme R-10, a mixture of pectinase, cellulase, and hemicellulose, is often utilized for plant-based foods, e.g., lettuce and radish, as well as seaweed. The enzyme α-amylase has been applied for the detection of NPs in noodles and wheat flour. One group developed a methanol extraction procedure for the determination of Au, CuO, and ZnO NPs in plant leaf materials (lettuce, corn, and kale) after an enzyme-based approach with Macerozyme R-10 caused changes in the recovered size distribution of CuONPs [42]. One article described the use of hydrogen peroxide digestion [34]. Following matrix degradation, further sample preparation steps can include filtration, settling, centrifugation, and dilution with surfactant-containing solutions. The influence of these steps on the obtained size distributions and recoveries is rarely investigated or discussed.

## 5. Analytical Approaches

Table 2 presents an overview of the main experimental parameters, calibration, and data analysis approaches described in the 49 selected papers reviewed herein. The references appear in the same order as Table 1.

With regard to the ICP-MS instrumentation platforms (Figure 3A), single quadrupole ICP-MS are the mass analyzers most widely used for single particle analysis in this emerging field (33 out of 49 papers) because of their comparatively low cost, higher robustness, and capability for fast NP detection. However, since single quadrupole instruments are sequential analyzers, only one *m*/*z* can be monitored at a time, limiting their multielemental detection and resolution. The capability of triple quadrupole technology allows for overcoming matrix interference in NP analysis and has been utilized in 12 out of the 49 selected papers, with half of them reporting the determination of metal oxides NPs, mostly TiO_2_, and the other half highlighting the analysis of AgNPs. Although double-focusing or sector field ICP-MS, when operating at low-resolution mode, can achieve higher sensitivity than quadrupole ICP-MS because of its geometry, this kind of instrumentation has been rarely used in the analysis of inorganic NPs in food to this date (two out of 49 papers). Interestingly, Noireaux et al. compared the performances of both high-resolution ICP-MS and triple quadrupole ICP-MS to characterize TiO_2_ NPs in different food products, concluding that double-focusing ICP-MS was able to detect smaller NPs than triple quadrupole ICP-MS [31]. While significant improvements developed within the last decade have increased the interest to revisit the use of time-of-flight ICP-MS for spICP-MS analysis, this trend has not translated yet into the analysis of inorganic NPs in food additives and food. In fact, only one study reported the use of time-of-flight ICP-MS technology for the quasi-simultaneous multi-elemental detection of Au, CuO, and ZnO NPs extracted from three different plant leaf materials (lettuce, corn, and kale) [42].

Typical sample introduction systems for the spICP-MS analysis of inorganic NP in food additives and food consists of a wide range of pneumatic nebulizers (Figure 3B) in combination with different spray chambers (Figure 3C) resulting in transport efficiencies in the range of 1% to 10% by applying sample flow rates of 0.1 mL min^−1^ to 1 mL min^−1^ (Table 2). Optimization of the operating conditions was largely unreported, with manual daily tuning for maximum sensitivity of the isotope(s) of interest carried out in a small fraction of the publications (12 out of 49) and autotune procedure reported in one case [55]. In spICP-MS, data acquisition was traditionally performed with millisecond dwell times in the past while a time resolution of hundreds of microseconds has been used more often in the last five to seven years. This trend is clearly manifested in the food analysis field where 27 papers reported the use of microsecond dwell times and the remaining 22 papers operated at millisecond dwell times (Figure 3D). The analysis time reported ranged from 60 s to 300 s, with 60 s being preferred by most of the laboratories (31 out of 49). As discussed above, the dominating use of sequential single quadrupole ICP-MS in this field clearly correlates with the fact that single element detection per analysis time was reported by 47 of the papers. Only two exceptions reported dual isotope detection [36] and simultaneous multi-elemental detection [42]. Rinsing procedures were highly unreported, with a sequence of diluted acid mixtures and surfactants or 2% nitric acid (*v*/*v*) being the most used approaches.

Overall, provided that there is suitable calibration of the instrument sensitivity, sample uptake rate, and transport efficiency (defined as the fraction of introduced sample that is transported to the plasma), spICP-MS allows the ability to obtain simultaneous information on both the number of NPs and the mass of the analyte per NP after a very short analysis time using off-the-shelf ICP-MS instruments. Calibration, sizing, and quantification strategies were summarized and described in detail in several reviews [17,18,19,20,21], so these factors will be only outlined below. Briefly speaking, when using pneumatic nebulizers, spICP-MS calibration is typically performed using NP standards of the same elemental composition and/or dissolved standard solutions of the element after taking into account the measure of the transport efficiency. This approach was reported in all 49 papers selected in this review.

With regard to the chemical composition of the NP used as calibration standards, AuNPs were reported in 43 papers and AgNPs in the remaining six papers. Monodispersed National Institute of Standards and Technology (NIST) Reference Material (RM) 8012 (nominal size 30 nm [70] and RM 8013 (nominal size 60 nm [71]) citrate-stabilized AuNPs have been widely adopted as calibration standards for spICP-MS in general, and for the spICP-MS analysis of inorganic NPs in food in particular. As illustrated in Figure 4A, NIST RM AuNPs were selected as calibration standards for 23 papers published between 2013 and 2021. Their prevalent use for spICP-MS calibration can be explained because of their well-defined and thoroughly characterized mean size, size distribution, and Au mass fraction, as well as their homogeneity and stability. Unfortunately, both NIST RM AuNPs have been out of stock since 2017. To date, the Quality Control Material LGCQC5050 Colloidal citrate-stabilized AuNPs (nominal diameter 30 nm), issued on February 2019 [72], has been used in one publication in this field [63]. In fact, the scarcity of appropriate NP RMs for spICP-MS calibration resulted in the wide use of commercially available NP suspensions of different sizes and coatings in 26 out of 49 papers (Figure 4A) for spICP-MS calibration. However, the accuracy of spICP-MS calibration based on commercial NPs can be compromised since value assignments provided by manufacturers are typically limited with respect to the number of NPs analyzed, just 100 NPs or so, and a more thorough in-house characterization is required [73]. Unfortunately, these required in-house characterization efforts for NP suspensions selected as calibration standards were rarely reported in the publications included in this review. Surprisingly, commercially available NPs, with very limited characterization information from the vendor, were often erroneously referred to as NP RMs, which are defined as materials, sufficiently homogeneous and stable with respect to one or more specified properties, which have been established to be fit for its intended use in a measurement process [74].

Two of the most popular approaches used in the spICP-MS analysis of inorganic NPs in food are to calculate transport efficiency, specifically employing the particle frequency and the particle size methods outlined by Pace et al. [75]. Both approaches require measurement of the sample flow, analysis of dissolved standard solutions (size method), and analysis of a standard NP suspension (i.e., NP RM) of known size and mass concentration or particle number concentration. It can be seen in Figure 4B that while the method for the calculation of the transport efficiency was not reported in four papers, the particle frequency method was the method of choice in 28 papers, followed by the particle size method in 13 papers.

Calculation of the transport efficiency using both methods was reported in four papers. Surprisingly, the dynamic mass flow, an indirect NP RM free method that solely relies on continuous mass measurements of the waste and sample uptake over time in a well-equilibrated ICP-MS, proposed by Cuello-Nuñez et al. in 2020 [76] has yet to be implemented in this field.

Due to the high time resolution, large data sets are generated in spICP-MS analysis and data reduction can be cumbersome. Aiming at automated data reduction, different strategies for increasing the sophistication of data processing have developed over the past decade, as illustrated in Figure 4C. For millisecond dwell time resolution, as particle events are detected as discrete pulses, datasets can be more easily handled by using simple algorithms implemented in spreadsheets such as the single particle calculation tool (SPCT) [77], used in nine papers, or spreadsheets developed in-house as employed in eight papers. For microsecond dwell time analysis, NPs are recorded as peaks that require more complex algorithms and software. Thus, proprietary software has been developed by most instrument manufacturers, allowing relatively easy processing of the acquired data; this is the preferred choice for data processing of spICP-MS analysis of NPs in food with a total of 25 papers. Alternatively, two different groups have also reported the development of custom scripts as data analysis tools in this field [42,44,67].

## 6. Method Validation

In this section, validation of spICP-MS for the analysis of inorganic NP in food additives and food will be discussed through a selection of classical analytical figures of merit. Thus, Table 3 includes detailed information (where applicable) about the limit of detection and quantification, repeatability, reproducibility, linear range, and trueness of spICP-MS in this arena. On Table 3, references appear in the same order as Table 1 and Table 2.

The size detection limit (LOD) was the figure of merit most often reported, for a total of 36 papers. While the size LOD is typically defined as the smallest ion burst which can be distinguished as a particle event, details on how it is mathematically calculated are not often provided. In spICP-MS, size LOD varies between NPs of different chemical compositions (i.e., its sensitivity, potential spectral interferences, and stoichiometry of NP) and the concentration of the dissolved analyte in the sample. Smallest-size LODs reported in this field ranged from 9 nm for Ag, 15 nm for Au, 18 nm for Se, 21 nm for HgSe, 26 nm for ZnO, 27 nm for TiO_2_, 35 nm for Fe_2_O_3_, 37 nm for Al_2_O_3_, 42 nm for CuO, 43 nm for Pb, and 89 nm for SiO_2_; these size LODs are comparable to values reported in general spICP-MS literature. In fact, measured NP diameters for smaller NP materials, reported in Table 1, are often very close to the reported size LODs, whose measured signals are at the method LOD and subject to high uncertainty. Interestingly, a clear correlation between smaller size LOD and the popularity of spICP-MS in measuring the size distribution of inorganic NPs, presented in Section 3, could not be established. Surprisingly, the size quantification limit (LOQ) was only reported by Waegeneers et al. for the analysis of AgNPs in confectionary [23].

One of the most important strengths of spICP-MS is its superb detection capability in terms of NP mass or number concentration comparable to NP concentrations in real-world environmental samples (on the order of ng L^−1^). However, mass concentration and particle number concentration LODs were reported to a lower extent with a total of 12 and 13 papers, respectively, with values in the ng L^−1^, and 10^5^ to 10^7^ L^−1^, respectively. Table 3 also shows the very limited information regarding the size linear dynamic range of NPs detectable in different food samples by spICP−MS, about one order of magnitude in particle diameter, reported in only two studies.

The assessment of the precision for the determination of particle size, mass concentration, and number concentration of inorganic NPs in food by spICP-MS was expressed in terms of repeatability and/or reproducibility, as displayed in Table 3. Repeatability for the measurement of particle size was reported in 14 papers and for mass and/or number concentration in 12 papers, while reproducibility was provided only in eight studies for particle size and four for mass concentration and/or number concentration. In short, results for the determination of median/mean particle diameter (1% to 10% repeatability standard deviation (SD), and 5% to 25% reproducibility SD) were more repeatable and reproducible compared to the determination of particle mass concentration or number concentration (2% to 47% repeatability SD and 7.5% to 90% reproducibility SD). It is important to highlight that the term reproducibility was erroneously used in the selected single laboratory studies. In fact, the intermediate precision was the actual indicator of precision, which is considered the most practically realizable estimate that can be achieved without evaluating a material in an interlaboratory study [78].

A summary of the trueness associated with the determination of particle size, mass concentration, and number concentrations, is also presented in the last two columns of Table 3. Information on the trueness of particle size measurements was largely reported, with a total of 40 papers, ranging from 60% to 116%. In practice, the most adopted approach for the evaluation of the trueness of particle size was through confirmation by Transmission Electron Microscopy (TEM) or Scanning Electron Microscopy (SEM) data either provided by the manufacturer, previously published, or acquired in-house. Alternatively, a comparison of electron microscopy results with spICP-MS size results of pristine NP suspensions used for spiking was performed in nine cases. The trueness of particle mass and/or number concentration was also largely reported, with a total of 32 papers, ranging from 12% to 127%. This larger bias compared to particle size can be understood because of the more challenging particle mass and number concentration measurements using spICP-MS (already discussed).

The comparison with expected concentration values, provided by manufacturers or RM producers, was the most used approach (15 papers), followed by the comparison with total concentration with conventional ICP-MS after acid digestion (12 papers).

While several in-house validation research projects have been developed at a single laboratory level [26,28,32,37,42,61], at this point it is necessary to highlight the publication of two international interlaboratory studies [24,62] that have established how well spICP-MS performs for the analysis of Ag and TiO_2_ NPs in food matrices. Under the framework of the NanoLyse project, Weigel et al. [62] organized the first interlaboratory comparison in this field for the size determination and quantification of AgNPs in chicken meat. In this interlaboratory study, results indicated greater variability in the particle number quantification than in the size characterization, yielding non-quantitative particle number concentration recoveries. These results could stem from numerous factors including the lack of stability of the NPs in initial suspensions and different matrices depending on the handling and storage conditions. Moreover, Geiss et al. [24] performed an interlaboratory comparison for the spICP-MS analysis of pristine TiO_2_ food additive E171, and in two types of confectionary products involving seven experienced participants. While spICP-MS resulted in significantly larger mean and median particle diameter than TEM, due to higher particle size LOD for spICP-MS and the difficulty of overcoming agglomeration in the sample preparation, the authors concluded that the study provided a good evaluation of spICP-MS practical validation.

## 7. Overview and Future Perspectives

This review identified several knowledge gaps in the field of spICP-MS analysis of NPs in food. In general, more screening studies are required to determine the background level of natural and incidental NPs in other food groups than seafood, e.g., fruits and vegetables. Single particle ICP-MS appears to be the ideal technique for it. The applicability of spICP-MS for the characterization of additional food additives other than titanium dioxide and silver should be investigated, e.g., iron oxides and hydroxides (E172) and aluminum (E173). Sample preparation procedures for foods with high starch and fat contents should be developed to investigate, e.g., the presence of inorganic NPs in bread or vegetable oils. The presence of NPs in feed and feed additives could be investigated with spICP-MS as well. Existing good practices of sampling that consider sample homogeneity and representativeness of subsamples should be applied for NP analysis by spICP-MS in the same way as for other methods.

Food additive SiO_2_ (used as an anti-clumping agent in powdered food products and a stabilizer in the production of beer ([79] accessed on 12 June 2023, [80] accessed on 12 June 2023)) is considered to be less toxic, especially at the allowable limit of 2% by weight; however, nanosized SiO_2_ may be created in the incorporation process and the toxicological effects then become unclear [80,81]. Hence, the toxicological effects of SiO_2_ are still under investigation. Research has been recently conducted toward the multi-technique characterization of food additive/food grade SiO_2_, with spICP-MS being employed as a main characterization avenue [82]. Although detection of Si is hampered heavily by isobaric interferences at *m*/*z* 28, this can be overcome through optimization of ICP-MS operating conditions (cold plasma and microsecond dwell time acquisition; Khan et al. (in preparation)) or switching to sector field ICP-mass spectrometers. Similar to the metal NPs discussed in this review, as more optimal spICP-MS methodologies present themselves toward the measurement and characterization of SiO_2_ in foods and food additive materials, more results can be gathered to inform regulatory decisions. This is also the case for the emergence of nano- and microplastics in the food industry.

Ideally, spICP-MS studies should always be supplemented with additional techniques to determine particle shape and composition. If this is not possible, limitations of spICP-MS should be communicated more clearly in publications when it comes to the assumptions that are made when determining particle size. Particle sizes could be presented as mass-equivalent sizes to highlight that the NP shape is either nonspherical or unknown. Considerations about the influence of NP composition and density on the NP size and mass concentration should be presented.

It is necessary to highlight that a general trend for using conventional spICP-MS experimental set-ups for the analysis of inorganic NPs in food additives and food is clearly prevalent. This can be explained due to the predominant application focus of the selected publications in this emerging field. It is expected to experience a gradual incorporation of alternative ICP-MS platforms, other than the single quadrupole instruments currently barely used in this field, over the course of the next years. Whereas higher sensitivity of double-focusing or sector field ICP-MS instruments will result in the lowering of the size detection limits, the trending use of time-of-flight ICP-MS will open the door to the identification, quantification, and classification of NPs of unknown sources based on their multielement fingerprint (i.e., engineered, incidental, and natural NPs).

A substantial collective effort should be made to report the optimization of the operating conditions and rinsing procedures that are largely unreported currently, to minimize this knowledge gap and to enable the transferability of measurement procedures more consistently across laboratories.

While the shortage of suitable NP RMs is not only an exclusive issue for spICP-MS but for any analytical technique for the characterization of NPs, it can be considered a major issue and current limitation affecting the accuracy of spICP-MS calibration, and sizing and quantification results of spICP-MS in general and in the food analysis field in particular. To overcome this lack of appropriate NP RMs, commercially available NP suspensions of different sizes and coatings are typically used instead. However, value assignments provided by the manufacturers have been demonstrated to be very limited with respect to the number of NPs analyzed, and therefore, a more thorough in-house characterization of commercial NP suspensions is required prior to their reliable use as calibration standards as recently outlined in [73,83]. This challenging task is currently considered an unexplored territory towards significant progress from the spICP-MS community in the food analysis field.

Unlike main experimental parameters, calibration, and data analysis approaches, information on the key analytical figures of merit for spICP-MS validation was generally unreported among all the selected studies. From the two interlaboratory studies and from the several in-house validation publications, two main remarks can be extracted with regard to spICP-MS validation in food analysis. While spICP-MS tends to perform reasonably well for the characterization of the central tendency of NP size distributions, important challenges remain in obtaining accurate and consistent particle number concentration measurements. The less than reliable number concentration results are related, but not limited to several factors: inaccurate calibration of transport efficiency, instability of NPs after extraction from food matrices, poor performance with regard to elemental sensitivity (leading to incorrect element responses factors), and loss of particles to the surface of the sample introduction system or the sidewalls walls of sample containers.

## Figures and Tables

**Figure 1 nanomaterials-13-02547-f001:**
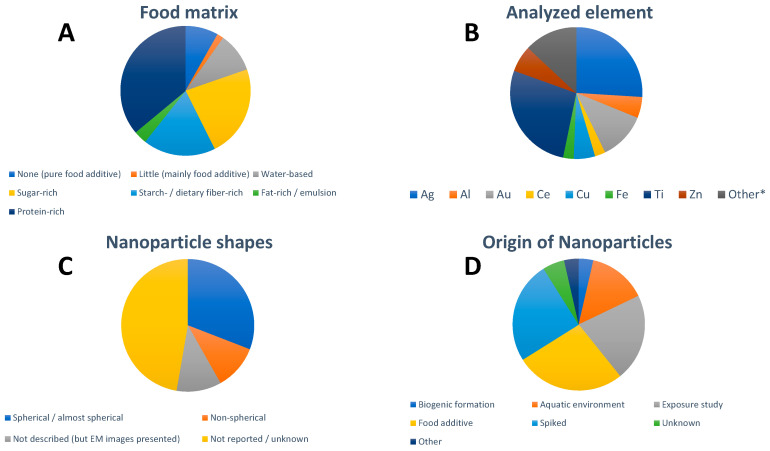
Overview of (**A**) studied food matrix, (**B**) analyzed element in the NPs, (**C**) NP shapes, and (**D**) origin of the studied NPs in the 49 identified studies focusing on spICP-MS analysis of food. * The designation “Other” refers to elements that were only studied in single publications, respectively (Hg, La, Nb, Pb, Pr, Pt, Se, Si, Y). For visual aid, the entries in the pie charts appear in order in a clockwise fashion following the legend beginning at the 12 o’clock position.

**Figure 2 nanomaterials-13-02547-f002:**
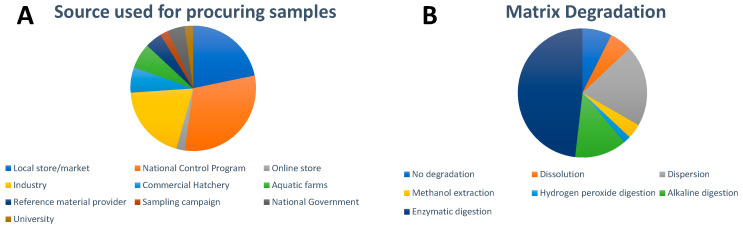
Overview of (**A**) sources used for procuring samples and (**B**) matrix degradation approach used for the spICP-MS analysis of inorganic NPs in food. For visual aid, the entries appear in order in a clockwise fashion following the legend beginning at the 12 o’clock position.

**Figure 3 nanomaterials-13-02547-f003:**
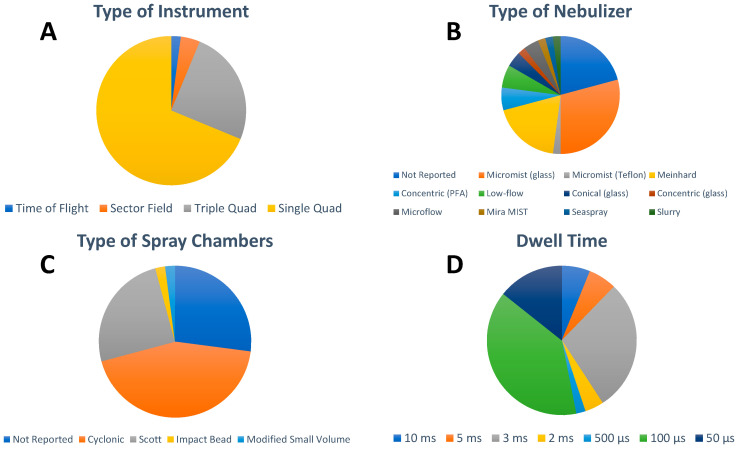
Overview of (**A**) type of instrument, (**B**) type of nebulizer, (**C**) type of spray chamber, and (**D**) detector dwell time used for the spICP-MS analysis of inorganic NPs in food. For visual aid, the entries in the pie charts appear in order in a clockwise fashion following the legend beginning at the 12 o’clock position.

**Figure 4 nanomaterials-13-02547-f004:**
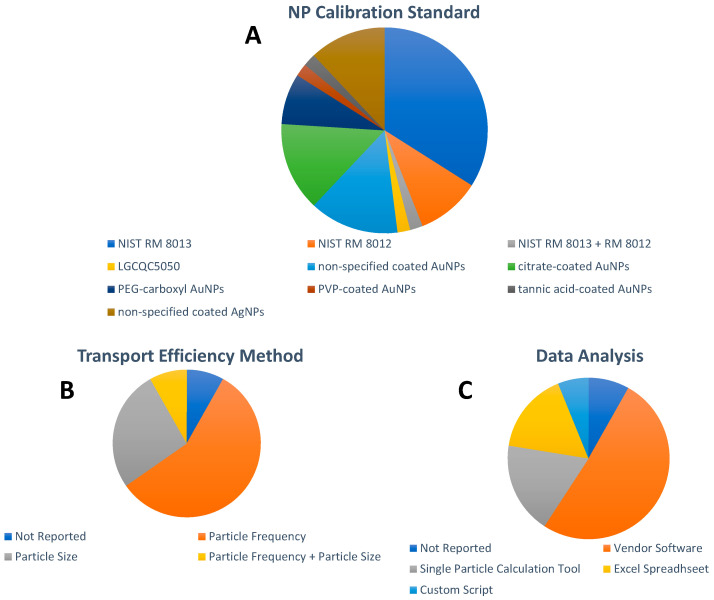
Overview of (**A**) NP calibration standards, (**B**) transport efficiency method, and (**C**): data analysis approach. For visual aid, the entries in the pie charts appear in order in a clockwise fashion following the legend beginning at the 12 o’clock position.

**Table 1 nanomaterials-13-02547-t001:** Overview of existing studies where spICP-MS is used to study NPs in food additives, food, and food-relevant matrices with focus on sample characteristics (including food matrix category, food matrix, analyzed element, assumed nanoparticle (NP) composition, reported NP shape, NP origin, and origin of the food sample), sampling (including homogenization approach, number/amount of samples analyzed), and sample preparation (including matrix degradation approach and further sample pre-treatment). Used abbreviations: BSA = bovine serum albumin, SDS = sodium dodecyl sulfate, TMAH = tetramethylammonium hydroxide (TMAH).

Food Matrix Category	Food Matrix (Only Eatable Parts)	Analyzed Element	Assumed NP Composition	Reported NP Shape	NP Origin	Origin of Food Sample	Homogenization Approach	Number/Amount of Samples Analyzed	Matrix Degradation Approach	Further Sample Pre-Treatment	Ref.
Pure food additives	-	Ag	Ag	Near spherical or irregular shape, aspect ratios from 1.07 to 1.42	Food additive (E174), powders (≤1 mm), flakes and petals (1 mm to 2 cm), leaves (>2 cm), 10 products in total	Online store	-	0.015 g	-	Ethanol wetting and dispersion in 0.05% (*m*/*v*) BSA by probe sonication	[3]
Pure food additives	-	Ag	Ag	Near spherical, aspect ratios from 1.07 to 1.28	Food additive (E174), 2-mm silver flakes and 8-cm silver leaves	Local store/online store	-	0.0154 g	-	Ethanol wetting and dispersion in 0.05% (*m*/*v*) BSA solution by probe sonication, vortex stirring, dilution with ultrapure water	[23]
Pure food additives	-	Ti	TiO_2_	-	Food additive (E171), 5 products	Provided by industry	-	50 mg	-	Dispersion in ultrapure water by bath sonication, filtration (1.2 μm cut-off)	[4]
Pure food additives	-	Ti	TiO_2_	Ellipsoidal	Food additive (E171)	Not specified (from collaborator)	-	40 mg	-	Dispersion in ultrapure water by probe sonication, dilution with ultrapure water	[24]
Pure food additives	-	Ti	TiO_2_	Fractal aggregate structure	Food additive (E171), 7 products	Local stores/online stores	-	(15.36 ± 0.10) mg	-	Ethanol wetting and dispersion in 0.05% (*m*/*v*) BSA solution by probe sonication	[25]
Pure food additives	-	Ti	TiO_2_	Near spherical, often agglomerated	Food additive (E171)	Online store	-	3.5 mg	-	Dispersion in ultrapure water by probe sonication, dilution with 4% nitric acid	[26]
Mainly food additive	Decoration dusting powders for silver metallic finishes (silver powder)	Ag Au Al	Ag Au Mica (KAl_2_[AlSi_3_O_10_](OH)_2_)	-	Food additive (E174) (E175) (E555)	Not specified (“commercially available”)	-	0.050 g	-	Dispersion in 0.1% (*v*/*v*) Tween 20, centrifugation, bath sonication, centrifugation to remove particles > 1 µm	[27]
Water-based	Apple and orange juice	Ag Au	Ag Au	Spherical	Spiked	Local store	-	1 mL	-	Dilution with ultrapure water	[28]
Water-based	Coffee with milk and espresso	Ag	Ag	Unknown	Unknown	Local coffee machine	Stirring with plastic spatula	100 mL	-	Dilution with ultrapure water	[14]
Water-based	Drink from vitamin (effervescent) tablets	Si	SiO_2_	Unknown	Unknown	Local store	-	1 vitamin tablet (3.8 g approx..)	Dissolution of the tablet in ultrapure water supported by vortexing	Dilution with ultrapure water	[14]
Water-based	Kefir (3.5% fat)	Se	Se	Spherical and ovoid	Biogenic formation (selenized bacterial strain spiked to food matrix)	-	-	-	Enzymatic digestion with proteinase XXIII	Dilution with 1% (*v*/*v*) methanol solution	[29]
Water-based	Milk	Ag	Ag	-	Spiked	Local store	-	5 g	Enzymatic digestion with Proteinase K	Dilution with ultrapure water	[30]
Water-based	Milk	Ti	TiO_2_	Almost spherical	Spiked	Local store	-	-	-	Dilution with ultrapure water	[31]
Water-based	Multifruit juice, white wine for cooking, hot chocolate, coffee, energy drink	Au	Au	Spherical	Spiked	Local store	Vigorous shaking	80 mL approx.	-	-	[32]
Sugar-rich	Cake decoration (edible golden stars)	Al Fe Si Ti	Al_2_O_3_ FeO(OH)·H_2_O SiO_2_ TiO_2_	Unknown	Unknown Unknown Food additive (E172) Food additive (E171)	Local store	-	1 golden star (65 mg approx.)	Dissolution of matrix in ultrapure water supported by sonication in high-intensity cup horn	Dilution with ultrapure water, vortexing	[14]
Sugar-rich	Cake decoration (inscription on plastic film)	Ti	TiO_2_	Almost spherical	Food additive (E171)	Local store	-	1 white inscription “Chocolat” scraped of the plastic film	Dissolution of matrix in 2 g/L sodium hexametaphosphate solution supported by sonication in high-intensity cup horn	Dilution with ultrapure water	[31]
Sugar-rich	Chewing gum (coating)	Al Si Ti	Al_2_O_3_ SiO_2_ TiO_2_	Unknown	Unknown Unknown Food additive (E171)	Local store	-	1 chewing gum	Dissolution of matrix in ultrapure water	Removal of gum base, dilution with ultrapure water	[14]
Sugar-rich	Chewing gum (coating)	Ti	TiO_2_	-	Food additive (E171)	Local store	-	1 chewing gum	Dissolution of matrix in ultrapure water supported by bath sonication	Dilution with ultrapure water, bath sonication	[33]
Sugar-rich	Chewing gum (coating), 2 products	Ti	TiO_2_	-	Food additive (E171)	Local store	Pooling of 3 chewing gums	3 chewing gums (from 4.3 g to 6.1 g)	Dissolution of matrix in ultrapure water supported by manual shaking	Removal of gum part, bath sonication, filtration (0.45 µm cut-off, dilution with ultrapure water	[32]
Sugar-rich	Chewing gum (coating)	Ti	TiO_2_	-	Food additive (E171)	Local store	Not detailed (representative subsample was prepared)	0.5 g	Heating with hydrogen peroxide just below boiling point, followed by evaporation	Dilution with 0.5% (*m*/*v*) BSA solution	[34]
Sugar-rich	Chewing gum (coating) and chocolate candy (sugar coating)	Ti	TiO_2_	Ellipsoidal	Food additive (E171)	Provided by industry (candies) or online store (chewing gum)	Pooling of six candies/three chewing gums	6 candies or 3 chewing gums	Dissolution of matrix in ultrapure water supported by manual shaking	Removal of gum/chocolate base, bath sonication, dilution with ultrapure water	[24]
Sugar-rich	Chewing gum (coating) and chocolate candy (sugar coating)	Ti	TiO_2_	Almost spherical	Food additive (E171)	Local store	-	1 chewing gum or 1 chocolate candy	Dissolution of matrix in ultrapure water	Removal of gum/chocolate base, dilution with ultrapure water	[31]
Sugar-rich	Chewing gums, chocolate candy, coated sweets, decorations on frozen desserts and pastries, 11 products in total	Ti	TiO_2_	-	Food additive (E171)	Local store	-	0.1 g to 2 g depending on sample	Dispersion in ultrapure water by sonication (for frozen desserts and pastries only specific part of sample taken for analysis)	Dilution with 0.1% (*v*/*v*) nitric acid	[35]
Sugar-rich	Chewing gum (coating), “typical French wedding hard candies” (almonds covered with sugar and E171 food coloring) Chocolate candy with peanut core (coating) Coconut syrup Soft candies with jelly center (wax glazing)	Ti	TiO_2_	-	Food additive (E171)	National control program	-	-	Dissolution of the matrix in ultrapure water supported by bath sonication	Centrifugation (1.2 µm cut-off), dilution with ultrapure water	[36]
Sugar-rich	Chocolate candy (sugar coating)	Ti	TiO_2_	Almost spherical	Food additive (E171)	Local store	Pooling of 8 candies	8 candies (around 7 g)	Dissolution of matrix in ultrapure water	Removal of chocolate core, sonication, filtration (0.45 μm cut-off)	[37]
Sugar-rich	Coffee creamer (powdered)	Al Si Ti	Al_2_O_3_ SiO_2_ TiO_2_	Unknown	Unknown Food additive (E551) Unknown	Online store	-	1 single serve packet	Dissolution of the matrix in hot coffee	Dilution with ultrapure water	[14]
Sugar-rich	Silvery coated candy beads	Ag Al	Ag Al	-	Food additive (E174) (E173)	Not specified (“commercially available”)	Pooling of 5 beads	5 beads (250 mg to 300 mg)	Dissolution of matrix in ultrapure water	Centrifugation to remove particles > 1 µm, dilution with ultrapure water, bath sonication	[27]
Sugar-rich	Silver coated chocolates (“sugar beans”) and silver pearls containing mainly sugar and wheat flour/corn meal	Ag	Ag	Near spherical, aspect ratios from 1.07 to 1.28	Food additive (E174)	Local store/online store	Pooling of three silver-coated chocolates/twelve silver pearls	3 items of silver-coated chocolates (~2 g) or 12 silver pearls (~2 g)	Dissolution in 0.05% (*m*/*v*) BSA solution supported by probe sonication; chocolate cores of silver-coated chocolates removed before probe sonication	Vortex stirring, dilution with ultrapure water	[23]
Sugar-rich	Silver pearls	Ag	Ag	-	Food additive (E174)	Local store	-	1 silver pearl	Dissolution of matrix in ultrapure water (only coating or whole pearl containing sugar core)	With and without filtration (0.45 µm cut-off), dilution with ultrapure water	[32]
Sugar-rich	Silver pearls (decoration of pastry)	Ag	Ag	Most NPs round, suggesting spherical 3D structure; presence of NPs with rectangular or triangular shape	Food additive (E174)	Local store	-	1 sugar pill (+/−180 mg)	Dissolution of matrix in ultrapure water	Dilution with ultrapure water	[38]
Sugar-rich	Silver pearls and silver-coated chocolates (silver beans), 10 products	Ag	Ag	Near spherical or irregular shape, aspect ratios from 1.07 to 1.42	Food additive (E174)	Local store	Pooling of 12 silver pearls/2–4 silver-coated chocolates	12 silver pearls or 2–4 silver-coated chocolates (about 2 g)	Dispersion in 0.05% (*m*/*v*) BSA solution by probe sonication	Remaining cores rinsed with 0.05% (*m*/*v*) BSA solution and removed (only for silver-coated chocolates), vortexing, dilution with ultrapure water	[3]
Sugar-rich	Sugar pearls	Ti	TiO_2_	-	Food additive (E171)	Provided by industry	Pooling of several pearls	1 g of pearls	Dissolution of matrix in ultrapure water supported by bath sonication	Filtration (1.2 µm cut-off)	[4]
Starch-/dietary fiber-rich	Chinese noodles (*n* = 21) and NIST SRM 1567a Wheat Flour	Al	Al_2_O_3_ or Al_2_Si_2_O_5_(OH)_4_ (kaolin)	Unknown	Unknown (different possibilities discussed)	National control program	Crushed in bag with rubber mallet followed by grounding in Retch centrifugal mill	50 g to 100 g	Enzymatic digestion with a-amylase	Dilution with ultrapure water	[39]
Starch-/dietary fiber-rich	Chinese noodles	Al Si	Al_2_O_3_ SiO_2_	Unknown	Unknown	National control program	Crushed in bag with rubber mallet followed by grounding in Retch centrifugal mill	30 mg	Enzymatic digestion with a-amylase	Hydrogen peroxide treatment supported by sonication (bath)	[14]
Starch-/dietary fiber-rich	Edible seaweed: Dulse and 5 sea lettuce	Ag	Ag	Spherical	Exposure study	Marine Research Station	Manual homogenization of seaweed samples; probe sonication of subsample with 2 mM/2 mM citric acid/trisodium citrate buffer (pH 4.5)	0.05 g	Enzymatic digestion with Macerozyme R-10	Filtration (cut-off 5.0 μm), dilution with 1.0% (*v*/*v*) glycerol	[40]
Starch-/dietary fiber-rich	Garden cress (shoots) and white mustard (leaves)	Pt	Pt	-	Exposure study	Grown from seeds	Manual grounding of dried plants using an agate mortar and pestle	0.025 g of dried and ground sample	Enzymatic digestion with Macerozyme R-10 after homogenization in citrate buffer using probe sonication	Filtration (0.45 μm cut-off), dilution with ultrapure water	[41]
Starch-/dietary fiber-rich	Kale, lettuce, and corn (leaves)	Au Cu Zn	Au CuO ZnO	Spherical - -	Exposure study and spiked Spiked Spiked	Local store (kale leaves) or grown from seeds (lettuce and corn)	Mechanical breakdown in homogenizer	1 g of leave material (fresh weight)	Two approaches: Enzymatic digestion with Macerozyme R-10 or methanol extraction after breakdown of leaf tissue with probe sonication; final method: methanol extraction	Enzymatic digestion: dilution with 0.5% (*v*/*v*) FL70, sonication, filtration (cut-off 5 µm); methanol extraction: dilution with 1% (*v*/*v*) Tween 80, sonication, filtration (cut-off 1 µm), dilution with deionized water	[42]
Starch-/dietary fiber-rich	Kale, lettuce, and collard green (leaves)	Cu	CuO	Primary NPs nearly spherical, while aggregates appear as rods and flakes	Exposure study	Local market	-	Circular pieces of leaf tissue (diameter = 6.35 mm)	Enzymatic digestion with Macerozyme R-10	Centrifugation, dilution with ultrapure water	[43]
Starch-/dietary fiber-rich	Lettuce (leaves)	Cu	CuO, Cu(OH)_2_	CuO: roughly spherical, Cu(OH)_2_: nanorods, commercial product Cu(OH)_2_: acicular	Exposure study	Grown from seeds	3 randomly selected plants homogenized in buffer at ratio of 1 g plant (fresh weight) to 20–200 mL buffer solution for a total homogenate volume 3.5 mL and 12 mL	0.5 mL of homogenate	Methanol extraction after breakdown of leaf tissue with indirect sonication (Vial Tweeter)	Dilution with 1% (*v*/*v*) Tween 80, sonication, filtration (cut-off 5 µm), dilution with ultrapure water	[44]
Starch-/dietary fiber-rich	Lettuce (leaves)	Zn	ZnO (only dissolved Zn taken up by plants)	-	Exposure study	Local store	Plants were lyophilized and the plant material ground in a mortar. Homogenization in citrate buffer using probe sonication	0.025 g of grounded sample	Enzymatic digestion with Macerozyme R-10	Filtration (0.45 µm cut-off)	[45]
Starch-/dietary fiber-rich	Mustard and lettuce plants (leaves)	Au	Au	Electron microscopy images presented, but shape not described	Exposure study	Agricultural university	Freeze-drying followed by grounding using a tissue homogenizer	(0.0250 ± 0.0003) g dried and ground plant tissue samples	Enzymatic digestion with Macerozyme R-10	Filtration (0.45 μm cut-off), dilution with ultrapure water	[46]
Starch-/dietary fiber-rich	Radish (roots)	Ce	CeO_2_	-	Exposure study	Grown from seeds	Freeze-drying and grinding with mortar	0.025 g	Enzymatic digestion with Macerozyme R-10 after homogenization by probe sonication	Settling for 15 min and filtration (cut-off 0.45 µm)	[47]
Starch-/dietary fiber-rich	Radish (roots)	Ti	TiO_2_	-	Exposure study	Grown from seeds	Freeze-drying and grinding with mortar	0.020 g	Enzymatic digestion with Macerozyme R-10 after homogenization with tissue grinder set	Settling for 60 min	[48]
Starch-/dietary fiber-rich	Wheat flour	Al Si	Al_2_O_3_ SiO_2_	Unknown	Unknown	Local store	-	30 mg	Enzymatic digestion with a-amylase	Hydrogen peroxide treatment supported by sonication (bath)	[14]
Fat-rich/emulsion	Salad dressing	Ti	TiO_2_	Unknown	Food additive (E171)	Local store	-	0.100 g	Dispersion in 0.1% (*m*/*m*) SDS solution by high intensity cup horn	Dilution with 0.1% (*m*/*m*) SDS, vortexing, dilution with ultrapure water	[14]
Fat-rich/emulsion	Sour cream (15% fat)	Se	Se	Spherical and ovoid	Biogenic formation (selenized bacterial strain spiked to food matrix)	-	-	-	Enzymatic digestion with proteinase XXIII	Dilution with 1% (*v*/*v*) methanol solution	[29]
Protein-rich	Aquatic organisms (invertebrates and fish): Taihu Lake shrimp, freshwater mussel, pearl mussel, Asian clam, snail, spiral shell; fish: stone moroko, yellow catfish, whitebait, crucian, carp, loach	Ag Ti	Ag Ti	-	From aquatic environment	Sampling campaign	Yes, but not described, followed by freeze drying	0.1 g of freeze dried sample	Alkaline digestion with TMAH	Settling (overnight), dilution with ultrapure water	[49]
Protein-rich	Bivalve mollusks: mussels, edible cockles, oysters, razor clams, variegated scallops, Atlantic surf clams, Japanese carpet-shell clams, grooved carpet shell	Ag	Ag	-	From aquatic environment	Local store	Mechanical blending (after byssus and/or shell were removed), 1 kg sample	1 g of homogenized sample	Enzymatic digestion with pancreatin and lipase, with and without simultaneous probe sonication	Centrifugation, dilution with 1% (*v*/*v*) glycerol solution, bath sonication	[50]
Protein-rich	Bivalve mollusks: various mussels	Ag Au Ti	Ag Au TiO_2_	Electron microscopy images presented, but shape not described	Spiked Spiked From aquatic environment	Aquaculture farm	Ground using tissue grinder and sonicated (before spiking), 60 mussels (35–45 g each); final method: freeze drying of ground samples	0.2 g to 0.5 g of wet sample, final method: 0.2 g of dry sample	Several approaches: Alkaline digestion with TMAH (2 protocols) or enzymatic digestion (5 protocols), final method: TMAH	Final method: centrifugation, dilution with ultrapure water	[51]
Protein-rich	Bivalve mollusks: oysters, mussels, scallops, clams, and ark shells	Ag Cu Ti Zn Au	Ag Cu (CuO for spiked) Ti (TiO_2_ for spiked) Zn (ZnO for spiked) Au	-	From aquatic environment Spiked	Offshore aquaculture farm	-	1.0 g of wet sample	Enzymatic digestion with pancreatin and lipase after probe sonication	Centrifugation, dilution with ultrapure water	[52]
Protein-rich	Bivalve mollusks: Asian clam	Ag Ti	Ag TiO_2_	Electron microscopy images presented, but shape not described	Exposure study and spiked	Sampling campaign	-	3 animals (whole soft body and specific tissues)	Enzymatic digestion with Proteinase K	Filtration (0.45 µm cut-off)	[53]
Protein-rich	Bivalve mollusks: Mussel NIST SRM 2976	Al Fe Si Ti	Al_2_O_3_ Fe_2_O_3_ SiO_2_ TiO_2_	Unknown	Unknown	NIST	Provided as a freeze-dried tissue powder	30 mg of freeze dried sample	Enzymatic digestion with Proteinase K	Dilution with ultrapure water	[14]
Protein-rich	Bivalve mollusks: clams and oysters	Au	Au	Electron microscopy images presented, but shape not described	Spiked	Sampling campaign	Not detailed (cut into pieces)	0.1 g of wet sample	Two approaches: Alkaline digestion TMAH supported by bath sonication in the beginning and enzymatic digestion with Protease K (excluded based on visual inspection of the samples)	Filtration (0.45 µm cut-off), dilution with 0.1% (*v*/*v*) Triton X-100 to TMAH concentrations of at least 1% (*v*/*v*)	[54]
Protein-rich	Bivalve mollusks: clams and oysters	Ce, La, Nd, Pr, Y (clams), Gd (oysters)	Ce, La, Nd, Pr, Y, Gd	-	From aquatic environment	Sampling campaign	Not detailed (cut into pieces)	0.1 g of wet sample	Two approaches: Alkaline digestion with TMAH supported by bath sonication in the beginning and enzymatic digestion with Protease K (excluded based on visual inspection of the samples)	Filtration (0.45 µm cut-off), dilution with 0.1% (*v*/*v*) Triton X-100 to TMAH concentrations of at least 1% (*v*/*v*)	[54]
Protein-rich	Bivalve mollusks: Mediterranean mussel	Ti	TiO_2_	Electron microscopy images presented, but shape not described	Exposure study	Mussel farm	Yes, but not described (25 mussels per exposure group without shells, 5 mussels per group with digestive glands and tissue prepared separately)	0.200 g subsample	Enzymatic digestion with Proteinase K	Dilution with ultrapure water	[55]
Protein-rich	Bivalve mollusks: mussels, edible cockles, oysters, razor clams, variegated scallops, Atlantic surf clams, Japanese carpet-shell clams, grooved carpet shell	Ti	TiO_2_	-	From aquatic environment	Local store	Mechanical blending (after byssus and/or shell were removed), 1 kg sample	1 g of homogenized sample	Enzymatic digestion with pancreatin and lipase, with and without simultaneous probe sonication	Centrifugation, dilution with 1% (*v*/*v*) glycerol solution, bath sonication	[56]
Protein-rich	Canned seafood (fish and bivalve mollusks): tuna, mackerel, anchovy, clam	Ag	Ag	-	From aquatic environment	Local store	Yes, but not described	3 batches for each brand of seafood, 0.25 g taken from each	Alkaline digestion with TMAH, bath sonication in the beginning of the digestion	Dilution with ultrapure water to 1% (*v*/*v*) TMAH concentration and 0.1% (*v*/*v*) Triton X-100, bath sonication	[57]
Protein-rich	Canned seafood (fish and bivalve mollusks): tuna, mackerel, anchovy, clam	Ti	TiO_2_	-	From aquatic environment	Local store	-	3 batches (cans) for each brand of seafood, 0.5 g of sample taken	Alkaline digestion using TMAH, bath sonication in the beginning of the digestion	Dilution with ultrapure water to 1% (*v*/*v*) TMAH concentration and 0.1% (*v*/*v*) Triton X-100, bath sonication	[58]
Protein-rich	Canned seafood (fish and bivalve mollusks): tuna, mackerel, anchovy, clam	Zn	ZnO	-	Unknown (different possibilities discussed)	Local store	Yes, but not described	0.25 g of sample	Alkaline digestion with TMAH, bath sonication in the beginning of the digestion	Dilution with ultrapure water to 1% (*v*/*v*) TMAH concentration and 0.1% (*v*/*v*) Triton X-100, bath sonication	[59]
Protein-rich	Chicken meat (lean, chicken breast)	Ag	Ag	-	Spiked	Local store	Paste by cryo-milling, vortexing after spiking	0.25 g	Enzymatic digestion with Proteinase K	Dilution with ultrapure water	[60]
Protein-rich	Chicken meat (lean, chicken breast)	Ag	Ag	Spherical	Spiked	Local store	0.2g subsample of meat cut into small pieces before spiking	0.2 g subsample	Enzymatic digestion with Proteinase K, tip sonication prior to addition of enzyme	Dilution with ultrapure water	[61]
Protein-rich	Chicken meat (lean, chicken breast)	Ag	Ag	Close-to-spherical	Spiked	Local store	Not detailed (described elsewhere)	Not detailed (described elsewhere)	Enzymatic digestion with protease K	Not detailed (described elsewhere)	[62]
Protein-rich	Chicken breast	Zn	ZnO	Polygonal shapes with curved ends	Migration study (ZnO NPs in polymer films)	Local market	-	1 g of chicken breast (cut into small pieces)	Aqueous extraction with Tris-HCl, supported by probe sonication	Dilution with ultrapure water	[63]
Protein-rich	Game meat (roe, deer, and wild boar)	Pb	Pb	-	Bullet fragments	National authorities	Production of slurry (homogenized tissue and water mixture)	17 to 35 g	Enzymatic digestion with Proteinase K	Dilution with ultrapure water	[64]
Protein-rich	Ground beef (93% lean)	Ag Au	Ag Au	-	Spiked	Local market	-	0.5 g	Alkaline digestion with TMAH, bath sonication for breaking down tissue and preventing particle aggregation prior to digestion, bath sonication in the beginning of the digestion	Diluting with ultrapure water to max. 1% (*v*/*v*) TMAH concentration	[65]
Protein-rich	Hen livers, muscles, kidneys, egg yolk and albumen	Ag	Ag	Spherical	Exposure study	Commercial egg-type hatchery	Yes, but not described	0.200 g subsample	Enzymatic digestion with Proteinase K	Dilution with ultrapure water	[66]
Protein-rich	Seafood (fish and mollusks): Tuna and swordfish, salmon and trout, horse mackerel and sardines, bream and flounder, other raw fishes (mainly bonito and yellowtail), squid and octopus, shellfish, and shrimp and crab	Hg	HgSe	Reference to other publication using high resolution TEM: spherical shaped, composed of 5-10 nm primary NPs	Biogenic formation	Local market	GM 200 knife mill to obtain a smooth paste (for fish internal organs removed before)	0.5 g of homogenized sample	Enzymatic digestion with pancreatin and lipase, supported by (ice-cooled) bath sonication in the beginning	Centrifugation, dilution with 1% (*m*/*v*) Tween-20, removal of dissolved Hg by repeated centrifugal ultrafiltration (50 kDa cut-off) using 1% (*v*/*v*) Tween-20 for washing and dilution	[67]
Protein-rich	Surimi (crab sticks, fresh and frozen products)	Ti	TiO_2_	Electron microscopy images presented, but shape not described	Food additive (E171)	Local store	Mechanical blending of 100 g approx.	1 g of homogenized sample	Enzymatic digestion with pancreatin and lipase	Centrifugation, dilution with 1% (*v*/*v*) glycerol solution, bath sonication	[68]

**Table 2 nanomaterials-13-02547-t002:** Overview of existing studies where spICP-MS is used to study NPs in food additives, food, and food-relevant matrices wit focus on experimental parameters (including instrument, sample introduction system, nebulizer, spray chamber, sample uptake, optimization of operating conditions, dwell time, analysis time, type of element/isotope detection, rinsing procedure), calibration (including type of used nanoparticle (NP) calibration standard, transport efficiency method), and data analysis. Used abbreviations: PEEK = polyether ether ketone, PEG = polyethylene glycol, PFA = perfluoroalkoxy, PVP = polyvinylpyrrolidone, SPCT = single particle calculation tool.

Instrument	Sample Introduction System	Nebulizer	Spray Chamber	Sample Uptake Rate (mL min^−1^)	Optimization Operating Conditions	Dwell Time	Analysis Time (s)	Element/Isotope Detection	Rinsing Procedure	NP Calibration Standard (Supplier)	Transport Efficiency Method	Data Analysis	Ref.
Triple quadrupole (Agilent 8800)	Peristaltic pump	Standard glass concentric nebulizer (Micromist)	Scott	0.47	-	3 ms	60	Single element	40 s rinse with HCl 5% (*v*/*v*) and a 160 s rinse with HNO_3_ 4% (*v*/*v*) or a 160 s rinse with a mixture of 1% (*v*/*v*) HCl (34% to 37%), 1% (*v*/*v*) HNO_3_ (67% to 69%), and 0.1% (*m*/*v*) Triton X-100	AuNPs of 30 nm (nanoComposix)	Particle frequency	-	[3]
Triple quadrupole (Agilent 8800)	Peristaltic pump	Standard glass concentric nebulizer (Micromist)	Scott	0.47	Manual tuning daily for the highest sensitivity	3 ms	60	Single element	40 s rinse with HCl 5% (*v*/*v*) and a 160 s rinse with HNO_3_ 4% (*v*/*v*) or a 160 s rinse with a mixture of 1% (*v*/*v*) HCl (34% to 37%), 1% (*v*/*v*) HNO_3_ (67% to 69%), and 0.1% (*m*/*v*) Triton X-100	Citrate-coated 30 nm AuNP (NIST RM 8012) and 30 nm AuNPs (nanoComposix)	Particle frequency	-	[23]
Single quadrupole (PerkinElmer NexION 300D)	SC Fast Peristaltic pump	Meinhard concentric glass nebulizer	Cyclonic	0.17	Manual tuning daily for the highest sensitivity	100 µs	60	Single element	-	Citrate-coated 60 nm AuNP (Sigma-Aldrich)	Particle size	Vendor software	[4]
Single and triple quadrupoles varied among the participants	Varied among the participants	Varied among the participants	Varied among the participants	0.17 to 0.47	Manual tuning daily for the highest sensitivity	100 µs to 3 ms	Various analysis times	Single element	-	Citrate-coated 60 nm AuNP (nanoComposix)	Particle size	Vendor software + SPCT	[24]
Single quadrupole (Thermo Scientific XSERIES 2)	Peristaltic pump + Autosampler	Burgener PEEK Mira Mist	Impact bead	-	Manual tuning daily for the highest sensitivity	3 ms	60	Single element	-	Citrate-coated 60 nm AuNP (NIST RM 8013)	Particle frequency	SPCT	[25]
Triple quadrupole (Agilent 8800)	-	Standard glass concentric nebulizer (Micromist)	Scott	0.47 ± 0.02	-	3 ms	60	Single element	Ultrapure water	AuNPs of 30 nm (nanoComposix)	Particle frequency	SPCT	[26]
Single quadrupole (PerkinElmer ELAN DRC-e)	-	Glass concentric Slurry nebulizer	Cyclonic	1.0	-	5 ms	60	Single element	-	PEG-carboxyl 100 nm AuNP (nanoComposix)	Particle size and particle frequency	Modified SPCT	[27]
Single quadrupole (PerkinElmer NexION 350D)	-	-	-	0.261	-	50 µs	60	Single element	-	Citrate-coated 60 nm AuNP (NIST RM 8013)	Particle size	In-house spreadsheet	[28]
Triple quadrupole (Agilent 8900)	Peristaltic pump	Standard glass concentric nebulizer (Micromist)	Scott	0.35	-	100 µs	60	Single element	10 s probe rinse with ultrapure water 60 s rinse with 1% (*v*/*v*) HCl, 1% (*v*/*v*) HNO_3_, and 1% (*v*/*v*) Triton X-100; 30 s rinse with 4% (*v*/*v*) HNO_3_; and 60 s rinse with ultrapure water	Citrate-coated 30 nm AuNP (NIST RM 8012)	Particle size	Vendor software	[14]
Single quadrupole (PerkinElmer NexION 350D) DRC mode: CH_4_ 0.4 mL min^−1^	Peristaltic pump	Meinhard concentric glass nebulizer	Cyclonic	0.33	-	100 µs	60	Single element	-	100 nm AgNPs (nanoComposix) 150 nm and 250 nm SeNPs (Sigma Aldrich) also tested	Particle frequency	Vendor software	[29]
Single quadrupole (Agilent 7900)	-	Standard glass concentric nebulizer (Micromist)	Scott	-	Manual tuning daily for the highest sensitivity	100 µs	60	Single element	-	40 nm AuNP (nanoComposix)	-	Vendor software	[30]
Sector Field (Thermo Scientific Element XR)	Self-aspiration	Glass concentric nebulizer (Seaspray)	Scott	0.31	-	5 ms	200	Single element	2 min wash in ultrapure water followed by a 3 min wash in 3% (*v*/*v*) HNO_3_ and a 2 min wash in ultrapure water	Citrate-coated 30 nm AuNP (NIST RM 8012)	Particle size	In-house spreadsheet	[31]
Single quadrupole (PerkinElmer NexION 300X)	Peristaltic pump	Meinhard glass nebulizer	Cyclonic	0.35	-	100 µs	100	Single element	-	AuNPs of 30, 50 and 100 nm (PerkinElmer)	Particle size	Vendor software	[32]
Triple quadrupole (Thermo Scientific iCAP TQ)	Peristaltic pump + Autosampler	Standard glass concentric nebulizer (Micromist)	Cyclonic at 2.7 °C	0.3	Manual tuning daily for the highest sensitivity	10 ms	180 or 300	Single element	-	Citrate-coated 30 nm AuNP (NIST RM 8012)	Particle frequency	Vendor software	[33]
Single quadrupole (Thermo Scientific XSERIES 2)	Peristaltic pump + Autosampler	Conical glass concentric	-	-	-	3 ms	60	Single element	-	Citrate-coated 60 nm AuNP (NIST RM 8013)	Particle frequency	SPCT	[34]
Single quadrupole (Thermo Scientific iCapQ)	Self-aspiration intake	Teflon Micromist	Cyclonic	0.36	-	10 ms	180	Single element	-	50 nm AuNP	Particle size and particle frequency	Modified SPCT	[35]
Single quadrupole (PerkinElmer NexION 300)	Peristaltic pump	Standard glass concentric nebulizer (Micromist)	Cyclonic at room temperature	0.4	-	100 µs	60 to 300	Dual isotope sequentially	-	Citrate-coated 60 nm AuNP (BBI)	Particle size and particle frequency	Vendor software	[36]
Single quadrupole (Agilent 7900)	Peristaltic pump	Standard glass concentric nebulizer (Micromist)	Scott	0.346	-	100 µs	100	Single element	-	Citrate-coated 60 nm AuNP (NIST RM 8013)	Particle size and particle frequency	Vendor software	[37]
Triple quadrupole (Agilent 8800)	-	Standard glass concentric nebulizer (Micromist)	Scott	0.5	-	3 ms	60	Single element	-	Citrate-coated 60 nm AuNP (NIST RM 8013)	Particle frequency	In-house spreadsheet	[38]
Single quadrupole (Thermo Scientific iCapQ)	Peristaltic pump	Low-flow concentric nebulizer	Cyclonic cooled	0.4	Manual tuning daily for the highest sensitivity	3 ms	60 to 180	Single element	-	Citrate-coated 60 nm AuNP (NIST RM 8013)	Particle frequency	In-house spreadsheet	[39]
Triple quadrupole (Agilent 8900)	Peristaltic pump	Standard glass concentric nebulizer (Micromist)	Scott	0.35	-	100 µs	60	Single element	10 s probe rinse with ultrapure water; 60 s rinse with 1% (*v*/*v*) HCl, 1% (*v*/*v*) HNO_3_, and 1% (*v*/*v*) Triton X-100; 60 s rinse with 4% (*v*/*v*) HNO_3_; and 60 s rinse with ultrapure water	Citrate-coated 30 nm AuNP (NIST RM 8012)	Particle size	Vendor software	[40]
Single quadrupole (PerkinElmer NexION 300X)	Peristaltic pump	Meinhard concentric glass nebulizer	Cyclonic	-	Manual tuning daily for the highest sensitivity	100 µs	100	Single element	-	Citrate-coated 30 nm and 60 nm AuNP (NIST RM 8012 and NIST RM 8013)	Particle size	Vendor software	[41]
Time of flight (TOFWERK *icp*TOF)	-	-	-	0.32 to 0.40	Manual tuning daily for the highest sensitivity	2 ms	-	Multiple elements	-	50 nm AuNP (Sigma-Aldrich)	Particle size	Custom script	[42]
Single quadrupole (Agilent 7900)	Peristaltic pump	Standard glass concentric nebulizer (Micromist)	Scott at 2 °C	0.346	-	100 µs	60	Single element	-	Citrate-coated 60 nm AuNP (nanoComposix)	-	Vendor software	[43]
Single quadrupole (Agilent 7900)	Syringe pump	DS-5 microflow concentric nebulizer	Modified small volume	-	-	100 µs	60	Single element	-	Citrate-coated 30, 60, 100, 200 nm AuNP (BBI)	Particle size	Custom script	[44]
Single quadrupole (Agilent 7900)	Peristaltic pump	-	-	-	-	100 µs	60	Single element	-	Citrate-coated 60 nm AuNP (NIST RM 8013)	Particle frequency	Vendor software	[45]
Single quadrupole (Agilent 7700x)	-	Standard glass concentric nebulizer (Micromist)	Scott	0.36	Manual tuning daily for the highest sensitivity	3 ms	60	Single element	HNO_3_ (1%, *v*/*v*), and ultrapure water	citrate-coated 60 nm AuNP (nanoComposix)	Particle size	In-house spreadsheet	[46]
Single quadrupole (Agilent 7900)	Peristaltic pump	-	-	0.35	-	100 µs	60	Single element	-	Citrate-coated 60 nm AuNP (NIST RM 8013)	Particle frequency	Vendor software	[47]
Triple quadrupole (Agilent 8900)	Peristaltic pump	-	-	0.35	Manual tuning daily for the highest sensitivity	100 µs	60	Single element	-	50 nm AuNP	Particle size	Vendor software	[48]
Triple quadrupole (PerkinElmer NexION 2000B)	-	-	-	0.32 to 0.36	-	50 µs	60	Single element	-	60 nm AgNP (J&K Scientific Ltd.)	Particle frequency	Vendor software	[49]
Single quadrupole (PerkinElmer NexION 300X)	Peristaltic pump + Autosampler	Concentric PFA nebulizer	Cyclonic	0.47	-	50 µs	100	Single element	-	Citrate-coated 60 nm AuNP (NIST RM 8013)	Particle frequency	Vendor software	[50]
Single quadrupole (PerkinElmer NexION 350D)	-	-	-	0.294	-	100 µs	60	Single element	-	PVP-coated 30, 60, and 100 nm Au NP (nanoComposix)	-	-	[51]
Single quadrupole (PerkinElmer NexION 350X)	-	-	-	0.29 to 0.32	-	100 µs	100	Single element	-	PEG-carboxil 30 nm, and 50 nm AuNP (nanoComposix)	Particle frequency	Vendor software	[52]
Triple quadrupole (Agilent 8900)	-	-	-	-	-	100 µs	60	Single element	-	Citrate-coated 60 nm AuNP (BBI)	-	Vendor software	[53]
Triple quadrupole (Agilent 8800)	-	Standard glass concentric nebulizer (Micromist)	-	0.346	-	3 ms	60	Single element	HNO_3_ (2%, *v*/*v*) and Triton X-100 (0.1%, *v*/*v*)	PEG-carboxil 50 nm AuNP (nanoComposix)	Particle frequency	Vendor software	[54]
Single quadrupole (Thermo Scientific iCapQ)	-	Microflow PFA-ST	Cyclonic	-	Autotune	3 ms	60	Single element	-	Citrate-coated 60 nm AuNP (NIST RM 8013)	Particle frequency	SPCT	[55]
Single quadrupole (PerkinElmer NexION 300X)	Peristaltic pump + Autosampler	Concentric glass nebulizer	Cyclonic	0.41 to 0.42	-	100 µs	100	Single element	-	Citrate-coated 60 nm AuNP (NIST RM 8013)	Particle frequency	Vendor software	[56]
Single quadrupole (PerkinElmer NexION 350D)	-	Meinhard concentric glass nebulizer	Cyclonic	0.26 to 0.28	-	50 µs	60	Single element	-	40 nm AgNP (PELCO^®^)	Particle frequency	Vendor software	[57]
Single quadrupole (PerkinElmer NexION 350D)	-	Meinhard concentric glass nebulizer	Cyclonic	0.26 to 0.28	-	50 µs	60	Single element	HNO_3_ (2%, *v*/*v*)	40 nm AgNP (PELCO^®^)	Particle frequency	Vendor software	[58]
Single quadrupole (PerkinElmer NexION 350D)	-	Meinhard concentric glass nebulizer	Cyclonic	0.26 to 0.28	-	50 µs	60	Single element	-	40 nm AgNP (PELCO^®^)	Particle frequency	Vendor software	[59]
Single quadrupole (Thermo Scientific iCapQ)	Peristaltic pump	Low-flow concentric nebulizer	Cyclonic cooled	0.4 mL	-	3 ms	180	Single element	-	Citrate-coated 60 nm AuNP (NIST RM 8013)	Particle frequency	In-house spreadsheet	[60]
Single quadrupole (Thermo Scientific XSERIES 2)	Peristaltic pump + Autosampler	Conical glass concentric	-	-	-	3 ms	60	Single element	-	Citrate-coated 60 nm AuNP (NIST RM 8013)	Particle frequency	SPCT	[61]
Varied among the participants	Varied among the participants	Varied among the participants	Varied among the participants	-	-	3 ms	60	Single element	-	Citrate-coated 60 nm AuNP (NIST RM 8013)	Particle frequency	SPCT	[62]
Single quadrupole (Agilent 7700x)	-	Standard glass concentric nebulizer (Micromist)	Scott	-	-	3 ms	60	Single element	-	Citrate-coated 30 nm AuNP (LGCQC5050)	Particle frequency	In-house spreadsheet	[63]
Single quadrupole (Thermo Scientific iCapQ)	Peristaltic pump	Low-flow concentric nebulizer	Cyclonic cooled	0.4	Manual tuning daily for the highest sensitivity	5 ms	60 to 180	Single element	Surfactant-containing acid mixture	Citrate-coated 60 nm AuNP (NIST RM 8013)	Particle frequency	In-house spreadsheet	[64]
Single quadrupole (PerkinElmer NexION 300Q)	-	Meinhard concentric glass nebulizer	Cyclonic	-	-	10 ms	200	Single element	-	Tannic acid-coated 100 nm AuNP (BBI)	Particle size	-	[65]
Sector field (Thermo Scientific Element2)	Peristaltic pump	-	-	1.0	-	2 ms	60	Single element	-	Citrate-coated 60 nm AuNP (NIST RM 8013)	Particle frequency	SPCT	[66]
Single quadrupole (Thermo Scientific iCapQ)	-	Concentric PFA	Cyclonic	-	-	0.5 ms	60 to 300	Single element	HNO_3_ (2%, *v*/*v*)	60 nm AgNP	Particle frequency	Custom script	[67]
Single quadrupole (PerkinElmer NexION 300X)	Peristaltic pump + Autosampler	Concentric PFA nebulizer	Cyclonic	0.41 to 0.43	-	100 µs	100	Single element	-	Citrate-coated 60 nm AuNP (NIST RM 8013)	Particle frequency	Vendor software	[68]

**Table 3 nanomaterials-13-02547-t003:** Overview of existing studies where spICP-MS is used to study NPs in food additives, food, and food-relevant matrices with focus on analytical figures of merit for method validation (including limit of detection (LOD) and limit of quantification (LOQ) for size, LOD/LOQ for mass and number concentration, size linear range, repeatability, reproducibility, trueness of size and concentration). Use abbreviations: AF4-ICP-MS = asymmetric flow-field flow fractionation coupled to inductively coupled plasma-mass spectrometry, AAS = atomic absorption spectroscopy, CLS = centrifugal liquid sedimentation, DLS = dynamic light scattering, SEM = Scanning electron microscopy, TEM = Transmission electron microscopy.

LOD/LOQ for Size	LOD/LOQ for Mass Concentration	LOD/LOQ for Number Concentration	Size Linear Range	Repeatability	Reproducibility	Trueness of Size	Trueness of Concentration	Ref.
LOD: 11 nm to 20 nm for Ag depending on the sample	-	-	-	-	-	Confirmed by TEM analysis	Comparison with ICP-MS after acid digestion	[3]
LOD: 9 nm for Ag LOQ: 11 nm to 13 nm for Ag	LOD: 0.1 ng g^−1^ for 110 nm AgNPs	LOD: (0.5 to 1.2) × 10^4^ mL^−1^	Assessed for ionic Ag concentration up to 2.5 ng mL^−1^	Size: 0.9% to 6.2% Mass Concentration: 16% to 29%	-	Confirmed by TEM analysis	Mass Concentration: 57% Number concentration: 51% Comparison with expected concentration	[23]
LOD: 40 nm for TiO_2_	-	-	-	-	-	Compared with TEM and CLS analysis	-	[4]
LOD: 38 nm for TiO_2_	-	-	38 nm to 475 nm for TiO_2_	Size: <10%	Size: <25%	Confirmed by TEM analysis	-	[24]
-	-	-	-	-	-	Comparison with SEM and AF4-ICP-MS analysis	Comparison with AF4-ICP-MS analysis	[25]
LOD: 39 nm for TiO_2_	LOD: 50 ng L^−1^ in diluted suspensions	LOD: 200 L^−1^ in diluted suspensions	-	Size: 4.9% Mass Concentration: 18% Number Concentration:17%	Size: 8.2% Mass Concentration: 23% Number Concentration:13% Expressed as intermediate precision	107% comparison with TEM analysis	Mass Concentration: 96% comparison with expected concentration	[26]
LOD ultrapure water: 25 nm for Au, 28 nm for Ag, 48 nm for Al LOD food matrix: 25 nm for Au, 31 nm for Ag, 81 nm for Al	-	LOD ultrapure water: 1.55 × 105 L^−1^ for Au, 1.50 × 105 L^−1^ for Ag, 2.32 × 105 L^−1^ for Al LOD food matrix: 2.21 × 105 L^−1^ for Au, 1.81 × 105 L^−1^ for Ag, 1.94 × 105 L^−1^ for Al	-	-	-	-	-	[27]
Particle calibration curve: 31 nm to 34 nm for Ag and Au Blank value: 0.6 nm to 2 nm for Ag and Au Dissolved calibration: 3 nm to 6 nm for Ag and Au	-	-	-	Size: 0.3% to 8.1% for Ag 0.4% to 47.0% for Au Expressed as within-day precision	Size: 1.9% to 25.3% for Ag 2.7% to 58.4% for Au Expressed as intermediate precision	93.2% to 114.3% for Ag 67.3% to 111.2% for Au Compared with TEM analysis supplied by the manufacturer	Number Concentration: 101.8% to 171.7% for Ag 87.0% to 119.4% for Au Comparison with expected concentration	[28]
LOD: 20 nm for Ag 37 nm to 52 nm for Al_2_O_3_ 35 nm to 76 nm for Fe_2_O_3_ 89 nm to 311 nm for SiO_2_ 37 nm to 43 nm for TiO_2_	-	-	-	-	-	Confirmed by SEM analysis	-	[14]
LOD: 18 nm for Se	-	LOD: 1.8 × 10^2^ mL^−1^ LOQ: 6.0 × 10^3^ mL^−1^	-	-		Confirmed by TEM analysis	-	[29]
-	LOD: 500 ng L^−1^ for 60 nm AuNPs	-	-	Size: (1 ± 1)% Mass Concentration: (2 ± 1)%	-		Mass Concentration: (97 ± 7)% (*n* = 3) Comparison with expected concentration	[30]
LOD: 32 nm for TiO_2_	-	-	-	-	-	74% Comparison with SEM analysis	Mass Concentration: 65% to 74% Comparison with ICP-MS after acid digestion	[31]
LOD: 18 nm for Au 32 nm for TiO_2_ > 200 nm for SiO_2_ 30 nm for Ag	LOD: 5 ng L^−1^ for 30 nm AuNPs 20 ng L^−1^ for 60 nm AuNPs 200 ng L^−1^ for 100 nm AuNPs 50 ng L^−1^ to 100 ng L^−1^ for TiO_2_ 100 ng L^−1^ for Ag	-	-	Mass Concentration: 1% to 5% for AuNPs of 30, 50, and 100 nm	Mass Concentration: 8% to 14% for AuNPs of 30, 50, and 100 nm	80% to 112% for AuNPs	-	[32]
LOD: 26 nm to 107 nm for TiO_2_	Background equivalent concentration: 0.024 µL L^−1^to 5.436 µL L^−1^	-	-	-	-	Agreement with published TEM	-	[33]
LOD: 20 nm to 50 nm for TiO_2_	-	-	60 nm to 300 nm for TiO_2_	-	-	Compared with SEM and AF4-ICP-MS analysis	Compared with AF4-ICP-MS analysis	[34]
LOD: < 30 nm for TiO_2_	-	-	-	-	-	-	-	[35]
LOD: 28 nm and 36 nm for ^48^TiO_2_; 67 nm and 85 nm for ^47^TiO_2_	-	-	-	-	-	-	-	[36]
LOD: 18 nm for Au 32 nm for TiO_2_	LOD: 5 ng L^−1^ for 30 nm AuNPs	-	-	Size: <6%	Size: <14%	Comparison with SEM, TEM, and DLS analysis supplied by the manufacturer	Comparison with ICP-MS after acid digestion	[37]
LOD: 13 nm for Ag	-	-	-	Mass Concentration: 38% RSD	-	-	Mass Concentration: 20% Comparison with expected concentration and with ICP-MS after acid digestion	[38]
LOD: 50 nm 50 to 60 nm for Al	-	-	-	Size: 2.7% Mass Concentration: 10.8%	-	-	Mass Concentration: 12% for noodle samples and 36.1% for SRM Wheat Flour. Comparison with ICP-MS after acid digestion	[39]
LOD: 14 nm for Ag	LOD: 2.2 ng g^−1^ LOQ: 7.7 ng g^−1^	LOD instrumental: 2.40 × 10^3^ mL^−1^ LOD sample: 4.34 × 10^7^ g^−1^	-	Size: <10% for size Number Concentration: 15% to 18%	-	Confirmed with TEM analysis	Number Concentration: (112 ± 12)% Comparison with expected concentration	[40]
-	-	-	-	-	-	-	-	[41]
LOD: 30 nm for Au, 42 nm for CuO, 62 nm to 78 nm for ZnO	-		-	-	Number Concentration: 12% to 25% for Au 7% to 17% for CuO 17% to 38% for ZnO	-	Mass Concentration: > 91% for Au 74% for CuO 68% for ZnO Comparison with ICP-MS after acid digestion	[42]
-	-	-	-	-	-	Confirmed by TEM analysis	-	[43]
-	-		-	-	-	Compared with spICP-MS analysis of pristine NPs	-	[44]
-	-	-	-	-	-	-	-	[45]
-	-	-	-	-	-	Compared with spICP-MS analysis of pristine NPs	Comparison with expected concentration	[46]
-	-	-	-	-	-	Compared with spICP-MS analysis of pristine NPs	-	[47]
-	-	-	-	-	-	Compared with spICP-MS analysis of pristine NPs	Compared with spICP-MS analysis of pristine NPs	[48]
LOD: 11 nm for Ag 27 nm for TiO_2_	-		-	-	-	-	Mass Concentration: 73.1% to 127% for Ag 5.5% to 23.3% for TiO_2_ Comparison with measured concentration of solution used for spiking	[49]
LOD: 13.6 nm to 16.2 nm for Ag	-	LOD: 0.417 × 10^7^ g^−1^	-	Number Concentration: 8%	-	Confirmed by SEM analysis	Number Concentration: 92% for 40 nm Ag 103% for 60 nm Ag Comparison with measured concentration of solution used for spiking	[50]
LOD: 15 nm to 17 nm for Au 10 nm to 12 nm for Ag 38 nm to 42 nm for TiO_2_	LOD: 0.005 ng g^−1^ for dissolved Au 0.005 ng g^−1^ for dissolved Ag 0.010 ng g^−1^ for dissolved Ti	-	-	-	-	Confirmed by TEM analysis	Number Concentration: (95 ± 1)% for Au (88 ± 0.9)% for Ag Comparison with expected concentration	[51]
-	-	-	-	-	-	(98.2 ± 3.8)% for 50 nm Au NPs in ultrapure water (106.8 ± 1.2)% for 50 nm Au NPs in enzyme-digested mollusk (97.2 ± 5.5)% for 60 nm for TiO_2_ NPs in ultrapure water (116.0 ± 4.4)% for 50 nm for CuO NPs in ultrapure water (104.5 ± 12.8)% for 80 nm for ZnO NPs in ultrapure water (94.0 ± 3.7)% for 80 nm for Ag NPs in ultrapure water Not significantly different from those measured in enzyme-digested mollusk and seawater Compared with expected nominal diameter	Number Concentration: (98.6 ± 3)% for 50 nm Au in ultrapure water (94.1 ± 10)% for 50 nm Au in enzyme-digested mollusk (91.0 ± 57)% for 60 nm for TiO_2_ in enzyme-digested mollusk (92.6 ± 7.1)% for 60 nm for TiO_2_ in seawater (81.2 ± 6)% for 50 nm for CuO in enzyme-digested mollusk (85.3 ± 8.4)% for 50 nm for CuO in seawater (78.1 ± 4)% for 80 nm for ZnO in enzyme-digested mollusk (80.6 ± 11.6)% for 80 nm for ZnO in seawater (85.3 ± 7)% for 80 nm for Ag in enzyme-digested mollusk (87.6 ± 8.9)% for 80 nm for Ag in seawater Not significantly different from those measured in enzyme-digested mollusk and seawater Compared with expected concentration	[52]
LOD: 16 nm to 20 nm for Ag	-	-	-	-	-	Compared with spICP-MS analysis of pristine NPs	-	[53]
-	-	-	-	-	-	95.9% for Au in alkaline treatment solution 103.6% for Au in enzyme treatment solution Compared with spICP-MS analysis of pristine NPs	Mass concentrations: 31.1 ng g^−1^ to 284.4 ng g^−1^ in clams 11.6 g g^−1^ to 127.3 ng g^−1^ for Y, La, Ce, Pr, and Nd in oysters. NPs of other elements not detected. Comparison with ICP-MS after acid digestion	[54]
LOD: 50 nm for TiO_2_	LOQ: 50 μg kg^−1^ for TiO_2_	-	-	Mass Concentration: 3% to 8% for TiO_2_	-	Confirmed by TEM analysis combined with EDX spectroscopy	Mass Concentration: 70% to 120% for TiO_2_ Comparison with ICP-MS after acid digestion	[55]
LOD: 24.4 nm to 30.4 nm for TiO_2_	Total Ti determination LOD: 31.7 ng g^−1^ LOQ: 105.6 ng g^−1^	LOD: 5.28 × 10^6^ g^−1^ LOQ: 1.76 × 10^7^ g^−1^	-	Size: 3% Number Concentration: 17%	-	Comparison with measured size of solution used for spiking	Number Concentration: 90% to 99% Comparison with measured concentration of solution used for spiking	[56]
LOD: 20 nm for Ag	-	LOD: 1.5 × 10^3^ mL^−1^ LOQ: 3.0 × 10^3^ mL^−1^	-	-	-	108.0% in water 110.5% in TMAH 1% Confirmed by TEM analysis supplied by the manufacturer	Number Concentration: 97.0% in water 88.4% in TMAH 1% Comparison with expected concentration	[57]
LOD: 35 nm for TiO_2_	-	LOD: 1.3 × 10^3^ mL^−1^	-	-	-	Confirmed by TEM analysis supplied by the manufacturer	Number Concentration: (97.6 ± 10.5)% in water (108.8 ± 7.2)% in TMAH 1% Comparison with expected concentration	[58]
LOD: 27 nm for ZnO	-	LOD: 3.0 × 10^5^ g^−1^	-	-	-	Confirmed by TEM analysis supplied by the manufacturer	Number Concentration: (84.7 ± 3.0)% Comparison with expected concentration	[59]
LOD: 15 nm for Ag	-	-	-	-	-	Comparison with TEM analysis	Mass Concentration: (68 ± 13)% Comparison with expected concentration	[60]
-	LOD: 0.05 mg kg^−1^	-	Assessed for 60 nm Ag concentration up to 50 mg kg^−1^	Size: 0.8% to 1.8% Mass Concentration: 6.7% to 11% Number Concentration: 6.4% to 14%	Size: 5.0% to 5.6% Mass Concentration: 8.9% to 16% Number Concentration: 7.5% to 18%	98% to 99% Compared with spICP-MS analysis of pristine NPs	Mass Concentration: 98% to 101% Number Concentration: 91% to 95% Compared with expected values	[61]
-	-	-	-	Size: 2% to 5% Number Concentration: 7% to 18%	Size: 15% to 25% Number Concentration: 70% to 90%	60% larger equivalent median diameter than the spiking solution using TEM and spICP-MS	Mass Concentration: 19% for Ag. Comparison with measured concentration of solution used for spiking by Neutron Activation Analysis	[62]
LOD: 26 nm for ZnO	-	-	-	-	-	Confirmed by TEM and DLS analysis	-	[63]
LOD: Wild boar: (56 ± 2) nm for Pb for 1 h of enzymatic digestion (80 ± 3) nm for Pb for 16 h of enzymatic digestion Roe deer: (46 ± 2) nm for Pb for 1 h of enzymatic digestion (43 ± 4) nm for Pb for 16 h of enzymatic digestion	LOD: 50 ng g^−1^ of dissolved Pb for wild boar 40 ng g^−1^ of dissolved Pb for roe deer	-	-	-	-	-	-	[64]
-	-	-	-	-	-	Comparison with expected size distribution of pristine NPs	Number Concentration: 90% to 95% for Au 83% to 106% for Ag Comparison with expected concentration Mass Concentration: 90% for Au 96% for Ag Comparison with ICP-MS after acid digestion	[65]
LOD: 10 nm for Ag	-	-	-	-	-	88% for Ag Confirmed by SEM-EDX analysis	Mass Concentration: 80% to 118% for Ag Comparison with AAS after acid digestion	[66]
LOD: 20.9 nm to 22.9 nm for HgSe	-	-	-	-	-	-	-	[67]
LOD: 31.3 nm to 37.1 nm for TiO_2_	-	LOD: 5.1 × 10^5^ g^−1^ LOQ: 1.7 × 10^6^ g^−1^	-	Size: 8% Number Concentration: 25%	-	Confirmed by TEM analysis	Number Concentration: (108 ± 5)% for 50 nm TiO_2_ (105 ± 4)% for 100 nm TiO_2_ Comparison with measured concentration of solution used for spiking	[68]

## Data Availability

Not applicable.

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
