# Peer review of "Application of Single Particle ICP-MS for the Determination of Inorganic Nanoparticles in Food Additives and Food: A Short Review"

_nanomaterials, 2023, doi:10.3390/nano13182547_

Round 1

Reviewer 1 Report

This work reviewed the single particle ICP-MS for detecting inorganic nanoparticles in food additives and food. The growing occurrence of NPs in food and food additives calls on high sensitive and reliable detection methods. The single particle ICP-MS is one promising technique to trace NPs. Thus, the topic is interesting. The review was not organized well and need to extensively revision. Current form are hardly to be accepted.

1. Title. Sample preparation should be included.

2. Introduction. The author mentioned FDA, EFSA. They are good. For the unbiased purpose, the food safety system in developing area should be add in.

3. The authors should add a section “Sampling” and “Sample preparation”, which are important for the reliable detection.

4. Section 3 and 4 shared the same section titles. Why?

5. Section 7. Challenges should be added.

Moderate editing of English language required

Author Response

This work reviewed the single particle ICP-MS for detecting inorganic nanoparticles in food additives and food. The growing occurrence of NPs in food and food additives calls on highly sensitive and reliable detection methods. The single particle ICP-MS is one promising technique to trace NPs. Thus, the topic is interesting. The review was not organized well and needs to be extensively revised. Current forms are hard to accept.

We respectfully disagree with this reviewer’s opinion regarding that an extensive revision is needed before it can be accepted. In fact, the perception of that this review manuscript is not organized well and needs to be extensively revised is not supported by the other three reviewers who highlighted that this is a "well-written review, easy to read with a good flow, and well structured".

  1. Title. Sample preparation should be included.

While we appreciate this suggestion, we believe that the current general title already includes all the aspects of the spICP-MS analysis in this field and is more appropriate for the scope of this review article. Thus, no changes have been made in this regard.

  1. Introduction. The author mentioned FDA, EFSA. They are good. For unbiased purpose, the food safety system in developing areas should be add in.

The introduction has been modified accordingly.

  1. The authors should add a section “Sampling” and “Sample preparation”, which are important for reliable detection.

We agree with the reviewer on the importance of sampling and sample preparation in our review. In fact, this corresponds to section 4. The title for that section has been modified accordingly.

  1. Sections 3 and 4 shared the same section titles. Why?

We apologize for this mistake. The correct title for section 4 is “Sample collection and sample preparation” and has been corrected as mentioned above.

  1. Section 7. Challenges should be added.

We believe that challenges in this field have been thoroughly presented in the manuscript. The paper focuses on numerous areas where spICP-MS research related to food and food additives could be improved by calling out the challenges in harmonizing results and also focusing on specific metrics and methodological details that are missing and/or were not explored in published literature. In fact, in the overview and future perspectives section, challenges related to the need for more studies investigating le preparation procedures for high starch and fat containing foods, inorganic NPs in breads and cooking oils, the shortage of NP reference materials, and the need for improvements in the reporting of figures of merit were highlighted.  

Reviewer 2 Report

This is a timely, comprehensive and well-written review. Single-particle ICP-MS is relatively little known to mainstream analytical chemists, so it is important to highlight the use of this technique in both food and nanoparticle analyses. My only concern is that Figures 1-4 are very difficult to interpret in black and white, they need to be in colour.

Author Response

This is a timely, comprehensive, and well-written review. Single-particle ICP-MS is relatively little known to mainstream analytical chemists, so it is important to highlight the use of this technique in both food and nanoparticle analyses. My only concern is that Figures 1-4 are very difficult to interpret in black and white, they need to be in colour.

We sincerely appreciate the very positive evaluation this reviewer made of our manuscript, including the recognition of the need for the present review. Figures 1 to 4 have been modified and are now displayed in color.

Reviewer 3 Report

In my opinion, the review manuscript is suitable for publication after minor revision. The topic is relevant and important for the field and the manuscript is written very well, easy to read with a good flow, with a large amount of relevant information related to the topic. The authors covered the topic comprehensively and in detail, covering all the important aspects, so the manuscript may become a starting point for researchers, experts and student involved in the field for their further work. As well, the authors addressed the current gaps and future perspectives in this topic, providing a critical opinion on top of the literature overview, which is welcomed.

There are some minor issues that need to be addressed:

The titles of tables and figures should be updated to be self-explanatory, meaning readers should be able to fully understand the contents of the tables and figures without the need to consult the main text. This also includes explanations of all the abbreviations.

L186: Studied instead of studies.

L485: The study of Khan et al. in preparation? Who is Khan, how can you cite someone else's work without it being published?

Author Response

In my opinion, the review manuscript is suitable for publication after minor revision. The topic is relevant and important for the field and the manuscript is written very well, easy to read with a good flow, with a large amount of relevant information related to the topic. The authors covered the topic comprehensively and in detail, covering all the important aspects, so the manuscript may become a starting point for researchers, experts and student involved in the field for their further work. As well, the authors addressed the current gaps and future perspectives in this topic, providing a critical opinion on top of the literature overview, which is welcomed.

We appreciate the positive evaluation of several aspects of this work, including the recognition of the value of this review for experts in the field and for a much broader audience.

There are some minor issues that need to be addressed:

The titles of tables and figures should be updated to be self-explanatory, meaning readers should be able to fully understand the contents of the tables and figures without the need to consult the main text. This also includes explanations of all the abbreviations.

The table and figure captions were modified accordingly, and more information added. Explanation of all abbreviations (except chemical symbols which are considered common knowledge) were included.

L186: Studied instead of studies.

Thanks for catching this typo. It has been fixed.

L485: The study of Khan et al. in preparation? Who is Khan, how can you cite someone else's work without it being published?

We appreciate the inquiry from the reviewer. Dr. Johnson and Dr. Montoro Bustos are coauthors on the Khan et al. Manuscript, and did extensive work toward the spICP-MS analysis of SiO2 food additive characterization using electron microscopy imaging and spICP-MS. This work has been presented at the SciX 2022 conference in Greater Cincinatti, Nothern Kentucky and at the 2023 PittCon Conference in Philadelphia. This project provides a real-world case and example of employing spICP-MS for future food additive work, but also highlights the challenges that can be addressed for the characterization of metal oxide nanoparticles in food. Additionally, Sadia et al. manuscript will be submitted to a journal this month.

Reviewer 4 Report

The manuscript “Application of Single Particle ICP-MS for the Determination of Inorganic Nanoparticles in Food Additives and Food: A Short Review” by Loeshner et al. is a compact overview of the application of sp-ICP-MS in the analysis of food products. It is a well structured collection of practical information on experimental parameters, but critically discusses also the limitations of the technique. One of the very valuable highlights is the analysis of the applied calibration standards and the discussion on the actual lack of appropriate reference materials. The structure of the manuscript fits its purpose, the English of the article is fine.

Specific comments:

Line 118. The authors describe only shortly, how de-agglomeration of particles could be achieved by probe sonication. However, probe sonication is known to have disadvantages like the possible release of particles from the head. I suggest to add some words on this and on the existence of other sonication methods (vial tweeter) to the text.

Line 223. “vegetative samples” sounds strange. Suggestion: plant vegetative part samples

Line 400. The authors collected the smallest LOD size reported for various materials, but do not comment on the differences between analytes and on the possible correlation between this and the popularity of sp-ICP in measuring the size distribution of these materials (lines 172-173).

Line 470 Typo: starchstarch

Lines 438 and 530 As the final message is about the challenge to improve the reliability of number concentration measurements, it would maybe deserve to extend the discussion on what is the reason for the much less reliable number concentration results. Is this somehow in correlation with size detection limit and neglecting the presence of smaller particles?

Author Response

The manuscript “Application of Single Particle ICP-MS for the Determination of Inorganic Nanoparticles in Food Additives and Food: A Short Review” by Loeshner et al. is a compact overview of the application of sp-ICP-MS in the analysis of food products. It is a well structured collection of practical information on experimental parameters, but critically discusses also the limitations of the technique. One of the very valuable highlights is the analysis of the applied calibration standards and the discussion on the actual lack of appropriate reference materials. The structure of the manuscript fits its purpose, the English of the article is fine.

We appreciate the positive evaluation of several aspects of this work, including the recognition of the practical value and critical discussion. The comments by this reviewer have encouraged us to clarify some aspects of this paper.

Specific comments:

Line 118. The authors describe only shortly, how de-agglomeration of particles could be achieved by probe sonication. However, probe sonication is known to have disadvantages like the possible release of particles from the head. I suggest to add some words on this and on the existence of other sonication methods (vial tweeter) to the text.

Additional information on the disadvantages of probe sonication has been added. The existence of other sonication methods has been now included.

Line 223. “vegetative samples” sounds strange. Suggestion: plant vegetative part samples

Thanks for the suggestion, the text has been modified accordingly.

Line 400. The authors collected the smallest LOD size reported for various materials, but do not comment on the differences between analytes and on the possible correlation between this and the popularity of sp-ICP in measuring the size distribution of these materials (lines 172-173).

We appreciate this helpful suggestion. A new sentence “Interestingly, a clear correlation between smaller size LOD and the popularity of spICP-MS in measuring the size distribution of inorganic NPs, presented in section 3, could not be established” has been added to the text.

Line 470 Typo: starchstarch

Thanks for catching the typo. It has been fixed.

Lines 438 and 530 As the final message is about the challenge to improve the reliability of number concentration measurements, it would maybe deserve to extend the discussion on what is the reason for the much less reliable number concentration results. Is this somehow in correlation with size detection limit and neglecting the presence of smaller particles?

Thanks. A new sentence has been added. “The less than reliable number concentration results are related, but not limited to several factors:  inaccurate calibration of transport efficiency, instability of NPs after extraction from food matrices, poor performance with regard to elemental sensitivity (leading to incorrect element responses factors), and loss of particles to the surface of the sample introduction system or the sidewalls walls of sample containers”.

Round 2

Reviewer 1 Report

This manuscript was revised well and can be published as it is.